# Repurposing conformational changes in ANL superfamily enzymes to rapidly generate biosensors for organic and amino acids

Jin Wang [1,2,3,4], Ning Xue [2,3,5], Wenjia Pan [2,4], Ran Tu [2,6], Shixin Li [2,5], Yue Zhang [2,4], Yufeng Mao [2,4], Ye Liu [2,4], Haijiao Cheng [2,4], Yanmei Guo [2,4], Wei Yuan [1,2,4], Xiaomeng Ni [2] & Meng Wang [1,2,4] ✉

Biosensors are powerful tools for detecting, real-time imaging, and quantifying molecules, but rapidly constructing diverse genetically encoded biosensors remains challenging. Here, we report a method to rapidly convert enzymes into genetically encoded circularly permuted fluorescent protein-based indicators to detect organic acids (GECFINDER). ANL superfamily enzymes undergo hinge-mediated ligand-coupling domain movement during catalysis. We introduce a circularly permuted fluorescent protein into enzymes hinges, converting ligand-induced conformational changes into significant fluorescence signal changes. We obtain 11 GECFINDERs for detecting phenylalanine, glutamic acid and other acids. GECFINDER-Phe3 and GECFINDER-Glu can efficiently and accurately quantify target molecules in biological samples in vitro. This method simplifies amino acid quantification without requiring complex equipment, potentially serving as point-of-care testing tools for clinical applications in low-resource environments. We also develop a GECFINDER-enabled droplet-based microfluidic high-throughput screening method for obtaining high-yield industrial strains. Our method provides a foundation for using enzymes as untapped blueprint resources for biosensor design, creation, and application.

Organic acids and amino acids are the critical basic building blocks of life as well as important intermediates and products in chemical manufacturing and the pharmaceutical industry[1]. The ability to quickly and precisely detect organic acids and amino acids in complex biological samples is significant for applications in medical diagnosis, environmental monitoring, and industrial microbial engineering[2–4]. Currently, the precise quantification of specific organic acids and amino acids relies on professional operations in specialized laboratories with sophisticated instruments such as high-performance liquid chromatography (HPLC), gas chromatography, and mass spectrometry (MS)[4–6]. However, such methods lack the detection throughput capacity and simplicity that are urgently needed in many applications, such as point-of-care testing (POCT) scenarios.

Transcription factor-based biosensors are the most widely used engineered genetically encoded small-molecule biosensors, but they still have many shortcomings. First, the time scales of metabolite production or transformation in vivo (~1 s) and transcriptional translation (~1–10 min) substantially differ[7], rendering real-time monitoring impractical for transcription factor-based biosensors. Second, significant amount of dedicated engineering work is required to make

[1]University of Chinese Academy of Sciences, 100049 Beijing, China. [2]Tianjin Institute of Industrial Biotechnology, Chinese Academy of Sciences, 300308 Tianjin, China. [3]Haihe Laboratory of Synthetic Biology, 300308 Tianjin, China. [4]Key Laboratory of Engineering Biology for Low-Carbon Manufacturing, 300308 Tianjin, China. [5]Tianjin University of Science & Technology, 300457 Tianjin, China. [6]College of Environmental and Resources, Chongqing Technology and Business University, 400067 Chongqing, China. ✉e-mail: wangmeng@tib.cas.cn

bacterial transcription factors compatible with eukaryotes due to the differences in their transcriptional processes[8]. Fluorescence Resonance Energy Transfer (FRET)-based biosensors generally perform better than transcription factor-based biosensors in terms of rapid signal response[9] and have been used for in vivo imaging and the real-time detection of various metabolites[10]. However, FRET-based biosensors typically have narrow dynamic ranges, limiting their ability to detect metabolites in vitro for screening purposes[11,12]. Compared with FRET-based biosensors, circularly permuted fluorescent protein (cpFP)-based biosensors usually have wider dynamic ranges (concerning the fluorescence change) and narrower excitation/emission wavelengths[13,14], making them suitable for cell biology applications, such as real-time imaging of intracellular metabolites[15].

Rapidly converting a suitable LBD into an applicable biosensor remains a challenge[16]. Moreover, the LBDs of conventional genetically encoded biosensors are almost always limited to naturally non-catalytic proteins, such as transcription factors, periplasmic binding proteins[17], and G protein-coupled receptors[18]. Enzymes that undergo significant conformational changes upon substrate/effector binding[19,20] represent an enormous untapped resource for genetically encoded biosensor generation. The ANL superfamily of ligases comprise three subfamilies: acyl- and aryl-coenzyme A (CoA) synthases, the adenylation domains (A domain) of the modular non-ribosomal

peptide synthases (NRPSs), and firefly luciferase[21]. ANL enzymes play important roles in fatty acid metabolism and transport, cell signaling, biofilm formation, antibiotics synthesis, and protein transport[22]. Members of the ANL superfamily have similar three-dimensional structures and catalytic reaction steps[21] and undergo large conformational changes during catalysis[23].

In this study, we develop a systematic method to design and generate genetically encoded circularly permuted fluorescent protein-based indicators to detect organic acids (GECFINDER) for the precise detection of various acids in complex biological samples (Fig. 1). By inserting circularly permuted enhanced green fluorescent protein (cpEGFP) into specific positions in ANL superfamily enzymes and performing high-throughput screening of small randomly mutated linker peptide libraries, we easily convert ANL superfamily enzymes into 10 biosensors for different acids by exploiting the conformational changes that occur during the catalytic processes. We then demonstrate that GECFINDERs can be applied for the rapid, and precise determination of amino acid concentrations in fermentation broth and human blood samples, providing a simple method for the in vitro accurate determination of amino acid concentrations in complex samples, which holds promise for POCT for metabolic diseases, such as infant phenylketonuria. In addition, we demonstrate that when used in combination with fluorescence-activated droplet sorting (FADS),

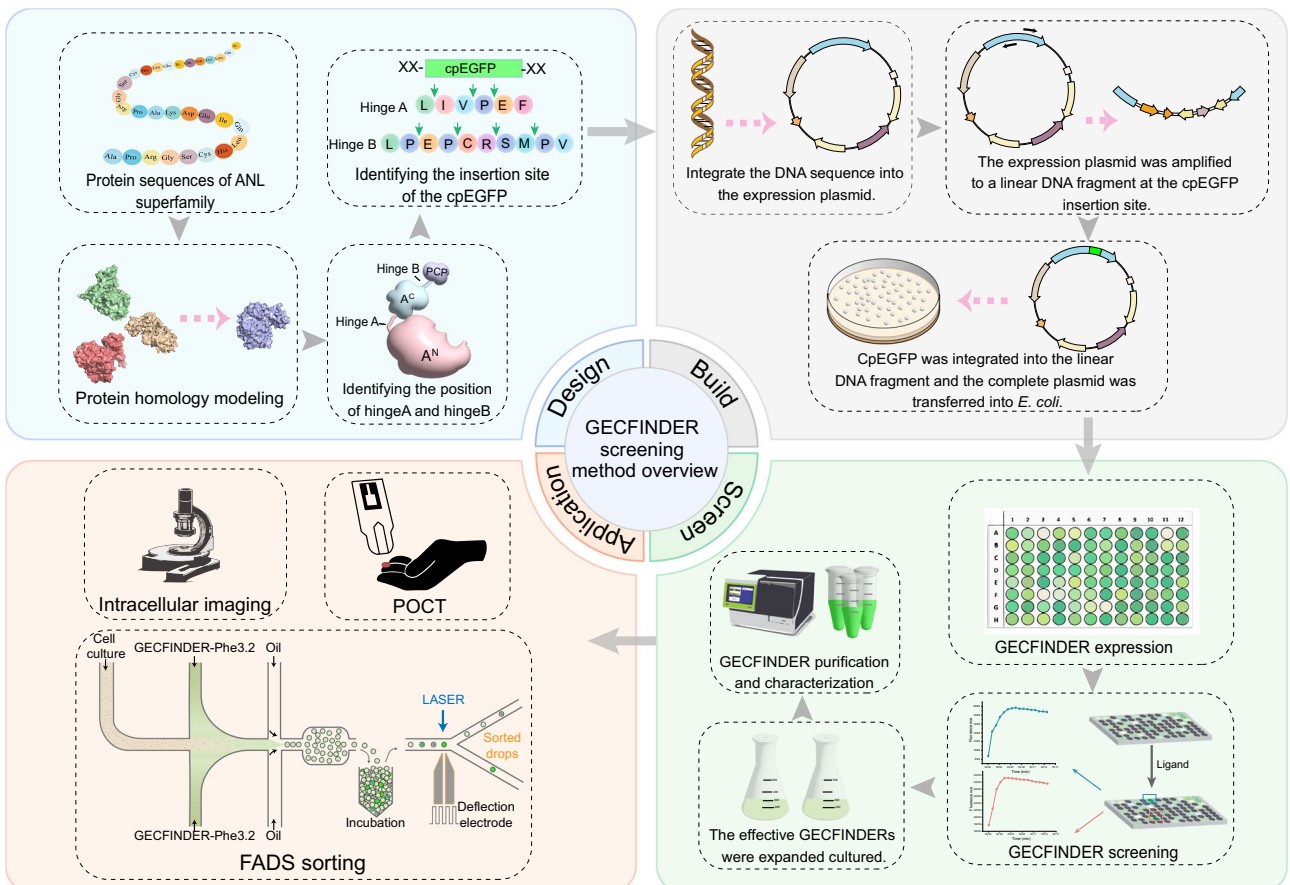

**Fig. 1 | The GECFINDER screening method overview.** The GECFINDER screening method can be roughly divided into 4 parts. The first part is the design of GECFINDER. After obtaining the ANL superfamily protein sequences, if there is no crystal structure, homology modeling is needed to determine the positions of hinge A and hinge B according to its three-dimensional structure, and cpEGFP with random linker is inserted into hinge A or hinge B. The second part is the construction of GECFINDER random linker library. Firstly, the DNA sequence of ANL superfamily member is integrated into the expression plasmid, and then random linker primers are designed at the insertion position of cpEGFP, and the linear DNA fragment of LBD is obtained

by PCR. The cpEGFP DNA fragment is integrated into the linear DNA fragments of LBD. The plasmid of GECFINDER random linker library is transferred into *Escherichia coli* to induce expression of GECFINDER random linker library. The third part is the screening of GECFINDER random linker library. The differences in fluorescence values of GECFINDER random linker library before and after the addition of ligand are detected by the microplate reader, and the effective GECFINDER is cultured, purified, and characterized. The fourth part is the various potential applications of GECFINDER including FADS screening, tissue cell imaging, POCT assay, etc.

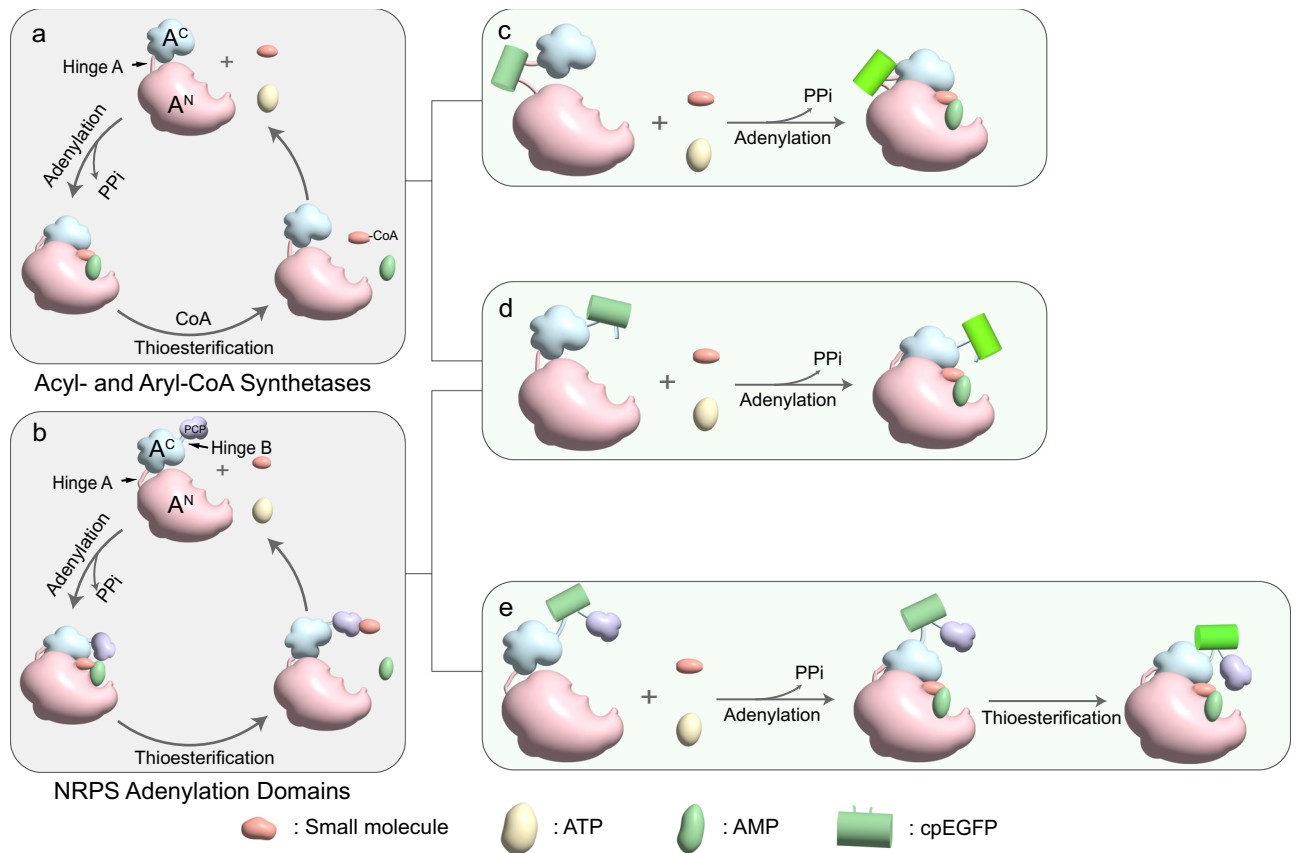

**Fig. 2 | Mechanism illustration of GECFINDER. a, b** Overview of the reactions catalyzed by acyl- and aryl-CoA synthetases and NRPS adenylation domains of the ANL superfamily. The hinge between the N-terminal large subunit and the C-terminal small subunit in the ANL superfamily is called hinge A, and the hinge between the C-terminal small subunit and the PCP domain is called hinge B. **c** cpEGFP is inserted into the hinge A between the N-terminal large subunit and the C-terminal small subunit of ANL superfamily members. When GECFINDER binds to its ligands and initiates the adenylation reaction, the N-terminal large subunit and C-terminal small subunit are close to each other, resulting in a change in the fluorescence intensity of cpEGFP. **d** cpEGFP is inserted into the end of the C-terminal small subunit of ANL superfamily members. When GECFINDER binds to its ligands and initiates the adenylation reaction, the N-terminal large subunit and C-terminal small subunit are close to each other, and the conformational changes at the C-terminal small subunit induces changes in the fluorescence intensity of cpEGFP. **e** cpEGFP is inserted into hinge B between the C-terminal small subunit of the ANL superfamily member and the PCP domain. When GEC-FINDER binds to its ligands and catalyzes the thioesterification reaction, the PCP domain is close to the C-terminal small subunit, and the conformational change of the C-terminal small subunit and the PCP domain leads to the change of the fluorescence intensity of cpEGFP.

GECFINDERs are excellent extracellular acids biosensors with significant application potential for high-throughput screening of high-yield industrial strains.

## Results

### Converting ANL superfamily enzymes into GECFINDERs

The catalytic process of the ANL superfamily enzymes starts in an open conformation, where the N-terminal large subunit and the C-terminal small subunit are separated by hinge A (Fig. 2a, b). After the substrate enters the binding pocket, adenosine triphosphate (ATP) hydrolysis is accompanied by the formation of aminoacyl-adenosine monophosphate (aminoacyl-AMP), resulting in a closed conformation. Subsequently, the active sites on the C-terminal subunit approach the substrate-binding pocket, and the activated substrate is transferred to peptidyl carrier protein (PCP) or CoA, completing the second thioester formation step. PCP or CoA then departs, completing the catalytic reaction, and the ANL superfamily enzyme returns to the open conformation, during which the C-terminal small subunit is displaced by approximately 140°[23] (Fig. 2a, b).

Alfermann et al. reported a FRET-based approach to study the conformational dynamics of NRPSs in the A PCP di-domain system. They generated FRET signals and monitored the conformational dynamics of NRPSs by removing four native cysteines from the A domain and introducing a single cysteine, chemically conjugated with Alexa Fluor 546 and fused enhanced green fluorescent protein at the end of the PCP domain[24]. This biosensor was not genetically encoded, which significantly narrowed its application scope. In addition, this strategy may not be directly applicable to other ANL superfamily members because the ANL superfamily enzyme shares less than 20% amino acid sequence identity with each other[25], and cysteine residues are not conserved. We thus speculated that FRET was probably not the best approach for transforming ANL superfamily enzymes into biosensors.

Members of ANL superfamily contains a large N-terminal subunit and a smaller C-terminal subunit connected by a highly flexible linker consisting of 5–10 amino acids (hinge A), and most NRPSs harbor a peptidyl carrier protein (PCP) domain connected by a 10–15 amino acid linker (hinge B) after the A domain (Fig. 2a, b). The flexibility of hinges A and B facilitates dramatic conformational rearrangement of ANL superfamily enzymes during catalytic reactions[26]. Based on the crystal structure of ANL family members, the regions where hinge A and hinge B were located appear relatively loose and flexible[27]. This flexibility may allow for the insertion of cpFP and transmit ligand coupled conformational changes to cpFP, resulting in changes to the fluorescent signal of cpFP. Therefore, we inserted cpEGFP with fixed short linkers at both ends (LE-cpEGFP-TR) between Leu439 and Ile440 in hinge A of *Nicotiana tabacum* 4CL (Nt4CL2, a member of ANL superfamily) and successfully transformed Nt4CL2 into a 4-coumaric

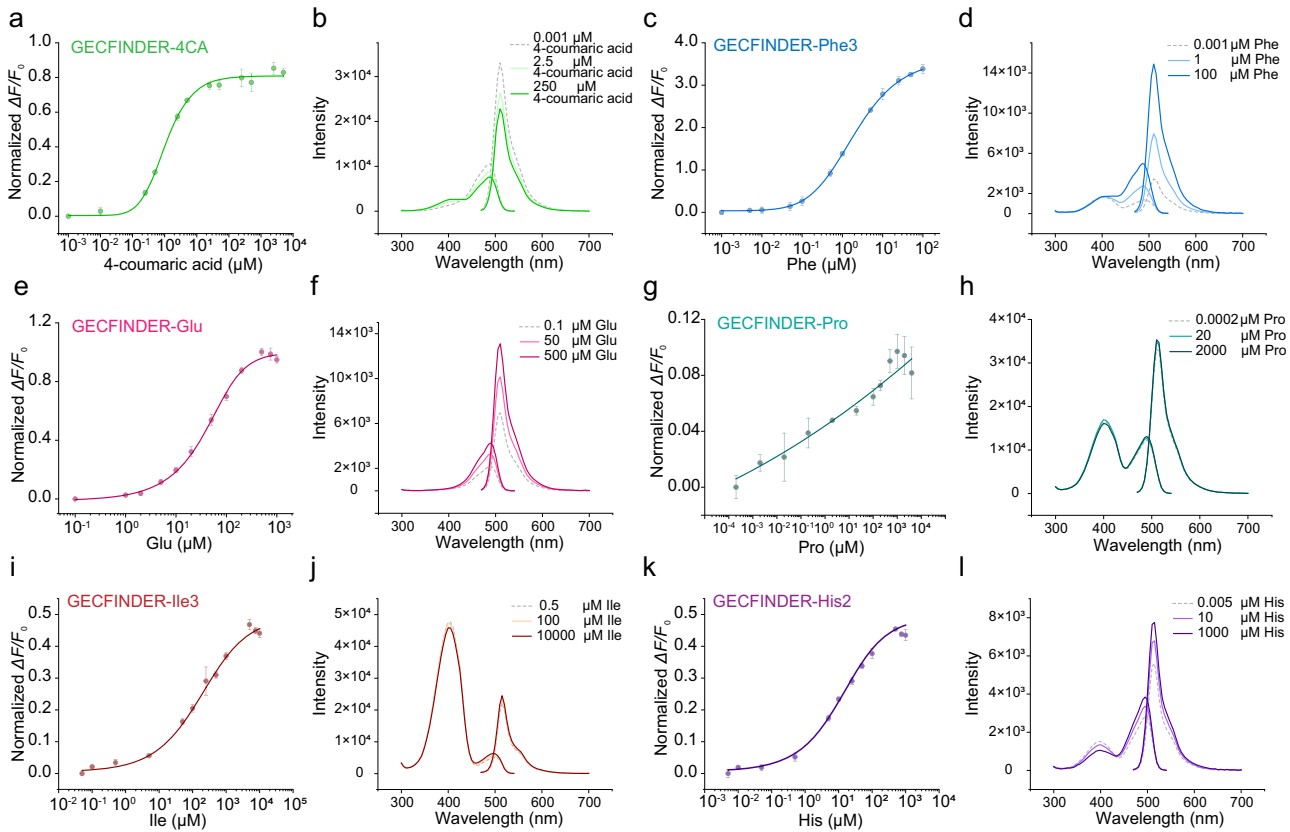

**Fig. 3 | Dose–response curves and fluorescence spectrums of GECFINDERs.**
**a, c, e, g, i, k** Dose–response curve of purified GECFINDERs with increasing concentrations of ligand in 100 mM Tris-HCl buffer (pH=7.5). The normalized $\Delta F/F_0$ value increased after ligand binding. All data shown are mean ± S.D. ($n=3$ biologically independent samples). **b, d, f, h, j, l** The fluorescence spectra of GEC-FINDERs changed as the ligand concentration increased. All data shown are mean ± S.D. ($n=3$ biologically independent samples). Source data are provided as a Source data file.

acid biosensor GECFINDER-4CA (Fig. 2c). Within the concentration range of 0.25–25 μM, the fluorescence intensity of GECFINDER-4CA increased with increasing 4-coumaric acid concentration, and the dose–response curve showed that the dynamic range ($\Delta F/F_0$) could reach 0.83 ± 0.03 (Fig. 3a, b). It should be noted that in Fig. 3a, the excitation wavelength was measured at 400 nm. However, the GECFINDER-4CA was ratiometric in excitation. The emission spectrum shown in Fig. 3b was measured when the excitation wavelength was 450 nm. As a result, it exhibited an opposite trend compared to the dose–response curve depicted in Fig. 3a.

Next, we inserted LE-cpEGFP-TR between Asp425 and Asp426 in hinge A of the glutamic acid-activated A domain of plipastatin synthetase (PpsA-GluA) (Fig. 2c). Surprisingly, the fluorescence intensity of the fused protein was not glutamate-dependent but ATP-dependent (Supplementary Fig. 1a–c). We speculated that this phenomenon was caused by the highly conserved Asp426 residue in hinge A, wherein the insertion of cpEGFP may disrupt the Glu A domain in the adenylation half-reaction. In subsequent experiments, we found that it was nearly impossible to generate effective biosensors by simply inserting cpEGFP with fixed linkers at both ends into hinge A or hinge B. Therefore, we inserted cpEGFP with random linkers (XX-cpEGFP-XX) into either hinge A (Fig. 2c) or hinge B (Fig. 2d, e) and screened the mutant library for effective biosensors. After optimizing the fluorescence screening parameters, a systematic screening method for the rapid generation of GECFINDERs was developed (Fig. 1).

We first applied our method to convert the phenylalanine-activated A domain from gramicidin S synthetase I (GrsA-PheA) into a biosensor. Four cpEGFP insertion sites were selected in hinge B of GrsA-PheA (Fig. 2e), and a total of 768 mutants were screened. Thirty-three mutants were found with $\Delta F/F_0 \geq 0.1$ among the three sites. To

study the effect of the linker on the performance of GECFINDER-Phes, three mutants from the same insertion sites were subjected to protein purification and characterization. Generally, the EC50 (half-maximal effective concentration) was similar among the three types of GECFINDER-Phes in the dose–response curves (Table 1), confirming that the affinity of GECFINDER to the ligand is not affected by the linker. However, the $\Delta F/F_0$ value significantly differ between the three types of GECFINDER-Phes, ranging from 0.27 ± 0.02 to 3.39 ± 0.11 (Fig. 3c, d and Supplementary Fig. 2a–d), indicating that the cpEGFP linker sequence strongly influenced the performance of GECFINDER. Next, we successfully generated four GECFINDERs for glutamic acid, proline, isoleucine, and histidine using the same screening method (Table 1, Fig. 3e–l, and Supplementary Fig. 2e–j). The cpEGFP insertion sites for all GECFINDERs and the detailed library screening data are reported in Supplementary Data 1.

## Substrate specificity engineering and cpEGFP insertion site optimization of GECFINDER

By comparing the sequences of the GrsA-PheA with a known structure (PDB ID: 1AMU) with 160 A domains, the 10 residues forming the A domain substrate-binding pocket were identified as the specificity-conferring code of NRPS A domain[28]. By using the leucine specificity-conferring code (Fig. 4a) as a guide, we were able to change the optimal substrate of GrsA-PheA from phenylalanine to leucine through the introduction of two mutations, T278M and A301G[28].

The ligand specificity of GECFINDER appears to depend solely on the substrate specificity of the original ANL superfamily enzyme. Because the LBD of GECFINDER-Phe is GrsA-PheA, the same mutation was generated in the corresponding position of GECFINDER-Phe3 (Fig. 4a), consequently switching the optimal ligand to leucine to

## Table 1 | Details of screened GECFINDERS

| Name | LBD | cpEGFP linker | Ex/Em, nm | Ligand | Operating range, µM | EC50, µM | ΔF/F$_O$ |
|---|---|---|---|---|---|---|---|
| GECFINDER-4CA | Nt4CL2 | LE-TR | 400/510 | 4-coumaric acid | 0.25–25 | 1.03 ± 0.06 | 0.83 ± 0.03 |
| GECFINDER-ATP | PpsA-GluA | LE-TR | 480/510 | ATP | 10–500 | 69.33 ± 4.69 | 0.49 ± 0.02 |
| GECFINDER-Phe1 | GrsA-PheA-PCP | CC-DR | 460/510[a] | Phe | 0.1–10 | 1.55 ± 0.23 | 0.27 ± 0.02 |
| GECFINDER-Phe2 | GrsA-PheA-PCP | CV-LL | 460/510[a] | Phe | 0.1–10 | 1.47 ± 0.15 | 0.62 ± 0.01 |
| GECFINDER-Phe3 | GrsA-PheA-PCP | VF-QS | 460/510[a] | Phe | 0.1–10 | 1.91 ± 0.08 | 3.39 ± 0.11 |
|  |  |  |  | Leu | ND[b] | ND[b] | 0.21 ± 0.01 |
|  |  |  |  | Tyr | 50–250 | 224.89 ± 17.87 | 1.24 ± 0.08 |
|  |  |  |  | S-β-Phe | 1–100 | 13.62 ± 1.28 | 0.75 ± 0.06 |
| GECFINDER-Glu | PpsA-GluA-PCP | HS-WL | 480/510 | Glu | 5–200 | 42.92 ± 3.93 | 0.95 ± 0.02 |
| GECFINDER-Pro | GrsB-ProA-PCP | PF-LH | 480/510 | Pro | ND[b] | ND[b] | 0.08 ± 0.02 |
| GECFINDER-Ile1 | BacA-IleA-PCP | VY-CA | 480/510 | Ile | 5–1000 | 160.95 ± 32.14 | 0.31 ± 0.01 |
| GECFINDER-Ile2 | BacA-IleA-PCP | RY-PG | 480/510 | Ile | 5–1000 | 221.35 ± 66.32 | 0.37 ± 0.01 |
| GECFINDER-Ile3 | BacA-IleA-PCP | VY-LS | 480/510 | Ile | 5–1000 | 186.46 ± 61.05 | 0.44 ± 0.01 |
| GECFINDER-His1 | BacC-HisA-PCP | GA-HC | 480/510 | His | 0.5–100 | 10.42 ± 4.74 | 0.35 ± 0.02 |
| GECFINDER-His2 | BacC-HisA-PCP | GG-PV | 480/510 | His | 0.5–100 | 13.25 ± 4.26 | 0.43 ± 0.02 |
| GECFINDER-Benzoic acid | SrCAR-PCP | GL-QR | 480/510 | Benzoic acid | ND[c] | ND[c] | 0.07 ± 0.01 |

[a]The excitation wavelength of GECFINDER-Phe is selected at the maximum ΔF/F$_O$, not the wavelength at the maximum fluorescence intensity.

[b]Due to the ΔF/F$_O$ values of GECFINDER-Phe3 for Leu and GECFINDER-Pro for Pro being less than 0.26, the signal-to-noise ratio was low, resulting in significant errors when fitting dose-response curves. As a result, the operating range and EC50 values for these sensors were not provided in this study.

[c]Not detectable. Additional parameters of the dose–response curves are provided in Supplementary Data 2.

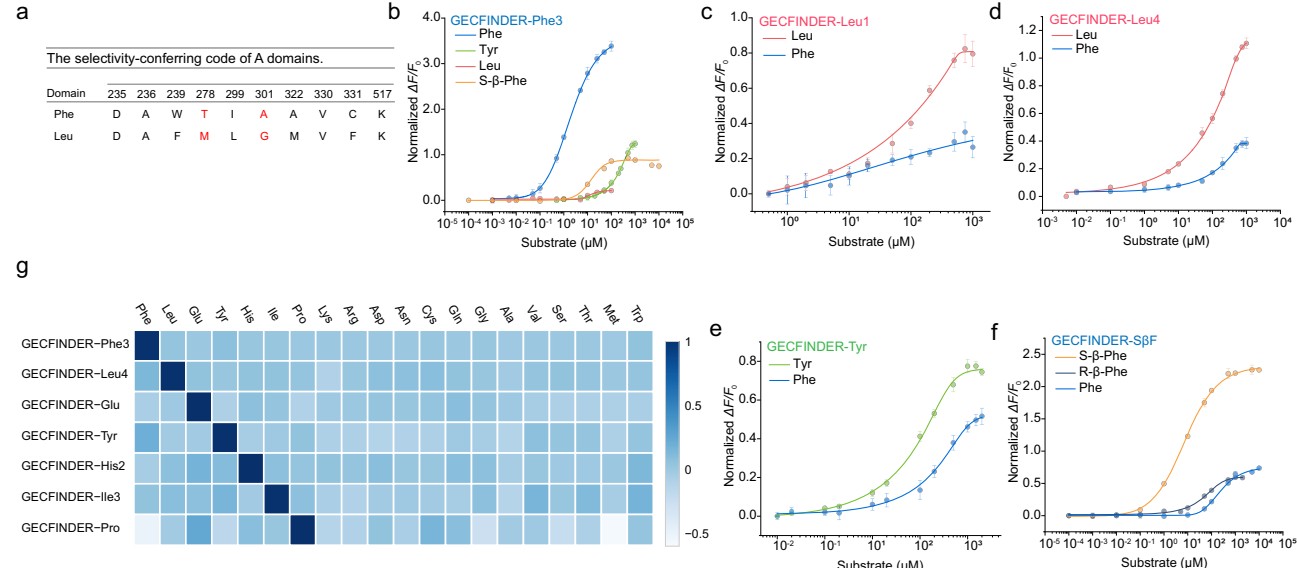

**Fig. 4 | Engineering ligand specificity of GrsA-PheA for generating GECFINDERs.**
**a** The selectivity-conferring code of Phe and Leu A domains. **b**−**f** Dose−response curve of purified GECFINDERs with increasing concentrations of ligands in 100 mM Tris-HCl buffer (pH=7.5). All data shown are means ± S.D. (*n* = 3 biologically independent samples). **g** A series of GECFINDER reactions was tested to show the GECFINDER's specificity with 20 proteinogenic amino acids. The values on the heatmap represent the ΔF/F$_O$ value normalized to each GECFINDER's maximal ΔF/F$_O$ value ((ΔF/F$_O$)/maximal ΔF/F$_O$). The concentrations of 20 amino acids were 100 µM. The color indicates the average value of three independent biological replicates after 15 min. All data shown are means ± S.D. (*n* = 3 biologically independent samples). Source data are provided as a Source data file.

generate GECFINDER-Leu1 (Table 2 and Fig. 4b, c). Furthermore, a computational structure-based redesign of GrsA-PheA was established to optimize the ligand-binding pocket of GrsA-PheA to enhance its leucine-binding ability[29]. GECFINDER-Leu2, 3, and 4 were constructed accordingly (Table 2, Fig. 4b, d, and Supplementary Fig. 3a, b). After a single-point mutation of W239S, the optimal substrate of GrsA-PheA was switched to tyrosine[30] to generate GECFINDER-Tyr (Table 2 and Fig. 4b, e).

Niquille et al. generated a mutant that changes the optimal substrate of TycA-PheA (phenylalanine-activated A domain of tyrocidine synthetase) from phenylalanine to S-β-phenylalanine using fluorescence-activated cell sorting (FACS)[31]. Because the protein sequences of TycA-PheA and GrsA-PheA are 62.2% homologous, we hypothesized that mutations targeting TycA-PheA might also apply to GrsA-PheA. Therefore, the same mutations (A236V, T238C-Δ329T-T330L-C331V) were introduced to the corresponding sites of GrsA-PheA. These mutations successfully changed the optimal ligand for GrsA-PheA to S-β-phenylalanine, generating GECFINDER-SβF (Fig. 4f). The EC50 value of GECFINDER-SβF for its original substrate, phenylalanine, was 234.25 ± 23.69 µM, representing a 123-fold reduction in

**Table 2 | Details of engineered GECFINDERs**

| Name | LBD | Mutation site(s) | Ligand | Operating range, µM | EC50, µM | ΔF/F$_O$ |
|---|---|---|---|---|---|---|
| GECFINDER-Leu1 | GrsA-PheA-PCP | T278M/A301G | Leu | 20–500 | 58.37 ± 24.78 | 0.80 ± 0.07 |
| | | | Phe | ND[a] | ND[a] | 0.26 ± 0.06 |
| GECFINDER-Leu2 | GrsA-PheA-PCP | I277L/T278L/A301G | Leu | 10–1000 | 207.42 ± 37.18 | 0.97 ± 0.03 |
| | | | Phe | 100–1000 | 605.54 ± 407.09 | 0.76 ± 0.10 |
| GECFINDER-Leu3 | GrsA-PheA-PCP | T278L/A301G/S447N | Leu | 10–1000 | 200.38 ± 7.76 | 2.33 ± 0.05 |
| | | | Phe | 10–1000 | 392.24 ± 51.79 | 1.46 ± 0.07 |
| GECFINDER-Leu4 | GrsA-PheA-PCP | I277L/T278L/A301G/S447N | Leu | 10–500 | 106.35 ± 24.46 | 1.10 ± 0.04 |
| | | | Phe | 50–200 | 157.69 ± 22.62 | 0.38 ± 0.04 |
| GECFINDER-Tyr | GrsA-PheA-PCP | W239S | Tyr | 10–500 | 89.81 ± 6.71 | 0.74 ± 0.02 |
| | | | Phe | 100–1000 | 258.00 ± 37.31 | 0.51 ± 0.04 |
| GECFINDER-SβF | GrsA-PheA-PCP | A236V/T238C/Δ329T/ T330L/C331V | Phe | 50–500 | 234.25 ± 23.69 | 0.74 ± 0.02 |
| | | | S-β-Phe | 0.1–100 | 7.40 ± 0.45 | 2.26 ± 0.02 |
| | | | R-β-Phe | 10–100 | 54.61 ± 7.66 | 0.59 ± 0.03 |

[a]Due to the ΔF/F$_O$ values of GECFINDER-Leu1 for Phe being less than 0.26, the signal-to-noise ratio was low, resulting in significant errors when fitting dose–response curves. As a result, the operating range and EC50 values for this sensor was not provided in this study. Additional parameters of the dose–response curves are provided in Supplementary Data 2.

EC50 compared with GECFINDER-Phe3 (Table 2). Moreover, the substrate selectivity (S-β-Phe EC50/R-β-Phe EC50) of GECFINDER-SβF was 0.14, indicating excellent stereoselectivity for S-β-Phe (Fig. 4b, f and Table 2). GECFINDER-Leus, GECFINDER-Tyr, and GECFINDER-sβF were generated by reprogramming the substrate specificity of GrsA-PheA using the malleability of the substrate-binding pocket of ANL superfamily members, demonstrating that switching the optimal ligand of a biosensor by redesigning its LBD was also a feasible strategy for constructing additional GECFINDERs.

Ligand specificity is an important biosensors feature. Generally, the NRPS A domains have high specificity for their substrates because they are the first "gatekeeper" of NRPSs and ensure that the correct monomers are activated and thioesterified[32,33]. Therefore, we speculated that this property would be inherited by our GECFINDERs because the ligand-binding pockets were unchanged during the GECFINDER creation process. Twenty proteinogenic amino acids were used to test the ligand specificity of GECFINDERs. As we expected, the GECFINDER-Phe/Glu/Pro/Ile/His had high specificity for their respective optimal ligands. GECFINDER-Phe3 had a EC50 value for its optimal substrate (phenylalanine) of only 1.91 ± 0.08 µM, and it exhibited a weak response to tyrosine and tryptophan (Table 1). The EC50 value of tyrosine was 224.89 ± 17.87 µM, more than 100-fold higher than that of phenylalanine (Fig. 4g and Table 1). The EC50 values of GECFINDER-Glu/Ile3/His2 for their corresponding optimal substrates were 42.92 ± 3.93, 186.46 ± 61.05, and 13.25 ± 4.26 µM, respectively, and no non-specific responses to other amino acids were observed (Fig. 4g and Table 1). The EC50 values for the corresponding optimal substrates of GECFINDER-Leu4, GECFINDER-Tyr, and GECFINDER-sβF obtained by binding-pocket engineering were 106.35 ± 24.46, 89.81 ± 6.71, and 7.4 ± 0.45 µM, respectively (Fig. 4g and Table 2). The substrate selectivities (optimal substrate EC50/Phe EC50) of GECFINDER-Leu4, GECFINDER-Tyr, and GECFINDER-sβF were 0.67, 0.35, and 0.032, respectively, exhibiting more than 70-fold reductions in their affinities for phenylalanine (Fig. 4b, d, e, f) and no non-specific responses (Fig. 4g and Supplementary Fig. 3c). We also investigated the dependence of GECFINDERs on Mg$^{2+}$ and ATP (Supplementary Fig. 4). All GECFINDERs exhibited no significant change in fluorescence intensity in the presence of Mg$^{2+}$ alone. Among them, the fluorescence intensity of GECFINDER-4CA is slightly reduced upon the presence of ATP and Mg$^{2+}$ under 400 nm excitation, while the addition of 4-coumaric acid enhances the fluorescence intensity (Supplementary Fig. 4a). This phenomenon corroborates with the distinct conformational changes observed in Hinge A of Nt4CL2 (Mg$^{2+}$+ATP) (PDB ID: 5BSM) and Nt4CL2 (Coumaroyl-AMP) (PDB ID:

5BST) reported in the literature (Supplementary Fig. 4k), where the direction of motion in hinge A differs[25]. GECFINDER-Phe3/Pro/Ile3/Leu4 exhibited a slight increase in fluorescence intensity when both Mg$^{2+}$ and ATP were present (Supplementary Fig. 4b, d, e, g). On the other hand, GECFINDER-Glu/His2/sβF/Tyr/Phe3.2 showed no significant change in fluorescence intensity when Mg$^{2+}$ and ATP were added (Supplementary Fig. 4c, f, h, i, j). During the determination of quantum yield, we observed that the absolute quantum yield of the GECFINDER variants ranged from 0.3 to 0.6 upon addition of the respective substrates, which show no significant difference with that reported in literature[34]. The specific values are provided in Supplementary Table 2.

As cpEGFP is sensitive to pH, we conducted a dose-response curve and fluorescence intensity analysis of GECFINDER-Phe3 at varying pH levels. We found that the fluorescence intensity of GECFINDER-Phe3 increased with increasing pH levels (Supplementary Fig. 3d), but ΔF/F$_O$ remained similar within the pH range of 6.5–7.5 (Supplementary Fig. 3e). However, at pH 8.5, ΔF/F$_O$ showed a slight decrease (Supplementary Fig. 3e). The EC50 value of GECFINDER-Phe3 remained similar at pH levels ranging from 6.5 to 8.5 (Supplementary Fig. 3e). These results demonstrate that GECFINDER-Phe3 is a reliable and stable biosensor that can be used in vitro under physiological conditions (pH 6.5–7.6). In addition, we expanded the range of ligand-binding domains (LBDs) applicable to GECFINDER and attempted optimization of insertion sites for cpEGFP with limited success. Please refer to Supplemental Note 1 for detailed information.

## Influence of cpEGFP insertion on the catalytic performance of ANL superfamily enzymes

Unlike traditional LBDs, ANL superfamily members are enzymes with catalytic functions. We thus investigated whether the insertion of cpEGFP into hinge A or hinge B disrupts the catalytic function of ANL superfamily members and whether the remaining catalytic function impairs the precision of GECFINDERs-based ligand concentration measurement. GECFINDER-4CA and GECFINDER-Phe3, with cpEGFP inserted in hinge A and hinge B, respectively, were selected as representative examples to determine whether the original two-step reaction occurred. First, the intermediates of the two GECFINDERs were measured to determine whether the first half-reaction occurred. Second, the final product 4-coumaroyl-CoA of GECFINDER-4CA and the protein molecular weight of GECFINDER-Phe3 before and after the reaction were measured to determine whether the second half-reaction occurred. For the first half-reaction, GECFINDER-4CA

reactions with and without CoA were analyzed by liquid chromatography–mass spectrometry (LC-MS). Compared to Nt4CL2, GECFINDER-4CA accumulated more of the intermediate product 4-coumaroyl-AMP in the absence of CoA. Firstly, the presence of the intermediate product indicated that the insertion of cpEGFP did not completely disrupt the first half-reaction of GECFINDER-4CA. Secondly, the accumulation of intermediate in GECFINDER-4CA may be due to the fact that the insertion of cpEGFP in Hinge A affects the folding of Nt4CL2 to some extent, making it easier for 4-coumaroyl-AMP to dissociate from the substrate binding pocket (Supplementary Fig. 5a, c). The surprising hydrolytic stability of acid anhydrides might be attributable to the resonance stabilization provided by the double bond at the ester linkage or the sequestration of the labile intermediate inside the active site and away from solvent or both[25]. The intermediate product of GECFINDER-Phe3 was detected by HPLC, and significantly higher AMP product peaks were observed in the GECFINDER-Phe3/ΔPCP and GrsA-PheA apo/holo (apo/holo: phosphopantetheine absence/presence on the PCP domain) reactions compared with the control (Supplementary Fig. 6a); however, the intermediate Phe-AMP was not observed. We hypothesized that this was because the majority of Phe-AMP was hydrolyzed to phenylalanine and AMP, and the remaining free Phe-AMP concentration was below the HPLC detection limit. This result confirmed that the insertion of cpEGFP into hinge B did not affect the first half-reaction of the A domain. For the second half-reaction, GECFINDER-4CA produced significantly less 4-coumaroyl-CoA than Nt4CL2 when CoA was present (Supplementary Fig. 5b, d), indicating that the insertion of cpEGFP significantly disrupted the second half-reaction. Electrospray ionization time-of-flight MS was used to measure the difference in molecular weight of GECFINDER-Phe3 and GrsA-holo before and after the reaction. The molecular weight of GrsA-holo after the reaction was 147 (M + 147) higher than that before the reaction (Supplementary Fig. 6b, c), whereas the molecular weight of GECFINDER-Phe3 did not change (Supplementary Fig. 6d, e). This result demonstrated that the second half-reaction did not take place in GECFINDER-Phe3 because the substrate was not covalently transferred onto the PCP domain following the first half-reaction.

In conclusion, whether inserted into hinge A or hinge B, cpEGFP did not affect the first adenylation step but significantly impacted the second thioesterification step. During the GECFINDER catalytic process, the substrate undergoes an adenylation reaction catalyzed by GECFINDER, producing an acid anhydride intermediate. However, the second-step thioesterification reaction is significantly inhibited, leading to hydrolysis of the labile intermediate into substrate and AMP. This process continuously consumes ATP, as it is required to sustain the catalytic cycle. While we did not find specific information on the $K_m$ value of ATP for Nt4CL2, a closely related homolog, At4CL2, has a reported ATP $K_m$ value of 0.163 mM[35]. Additionally, the GrsA-Phe A domain exhibits a $K_m$ value of 0.15 mM for ATP[36]. Therefore, we adding sufficient ATP[37] to the reaction mixture to maintain ATP concentration at saturation levels for long times in all experiments to avoid any potential impact on the measurement results of GECFINDER.

## The accurate determination of amino acid concentration in biological samples using GECFINDERs

Transcription factor biosensors can be used as screening tools for high-throughput screening of amino acids producing strain[38], but they are rarely used for rapid and accurate in vitro quantification of amino acids. The precise quantification of amino acids relies on techniques such as HPLC, LC-MS, and MS[39], which have limited throughput. As genetically encoded biosensors that can be applied in vitro, GECFINDER may provide a simple, rapid, and high-throughput method for the precise quantification of amino acids. In our study, we used GECFINDER-Glu to quantify the concentration of glutamic acid in fermentation broth samples of glutamate-producing strains. The glutamic acid concentration measured using GECFINDER-Glu are consistent with those obtained using an SBA-40D biosensor analyzer (Supplementary Fig. 7a, b). In addition, GECFINDER-Glu can simultaneously measure 96 samples in a microwell plate within 5 min, while SBA can only detect one sample at a time, and it takes at least 100 min to detect 96 samples.

Inborn errors of metabolisms (IEMs) are a group of complex monogenic disorders that result from errors in the genetic code, leading to reduced or deficient enzyme activity in a single metabolic pathway[40]. Regular monitoring of metabolite concentrations is necessary for many IEM patients to prevent harmful aberrant metabolite concentrations in the body. POCT methods, which offer inexpensive and portable alternatives to large specialized equipment, have potential for newborn screening in less economically developed areas[41]. To this end, GECFINDER-Phe3 could potentially serve as an affordable POCT method for IEM newborn screening. In this study, we evaluated the accuracy of GECFINDER-Phe3 in measuring phenylalanine, a diagnostic marker for phenylketonuria, in human serum samples. We compared the results of GECFINDER-Phe3 to those obtained using HPLC by simultaneously determining the concentration of phenylalanine in human serum samples (Supplementary Fig. 7c, d). Our results demonstrate that GECFINDER-Phe3 and HPLC provide consistent results (Supplementary Fig. 7e). Overall, we used GECFINDER-Glu and GECFINDER-Phe3 to precisely determine the concentrations of glutamic acid and phenylalanine, respectively, in complex biological samples. Compared to SBA and HPLC, GECFINDER does not require complex equipment, and there is no need to use chromatographic columns, pre-column or post-column derivatization, which significantly simplifies the amino acid determination process. GECFINDER is a high-throughput method, allowing for rapid and simultaneous determination of multiple samples. This is particularly advantageous in clinical applications, where speed and accuracy are essential. GECFINDER can be developed as a kit for rapid quantification of amino acid concentrations in biological samples or as a rapid diagnostic tool for POCT, which could be used in clinical settings or field conditions. In summary, GECFINDER offers several advantages over traditional methods of amino acid determination. It is easy to use, highly sensitive and specific, and allows for high-throughput analysis. These features make GECFINDER a promising tool for studying metabolism, screening compounds, and clinical applications such as disease diagnosis and monitoring.

## Development of a droplet-based microfluidic high-throughput screening method using GECFINDERs

Currently, genetically encoded biosensors based on the cpFP have been primarily employed for real-time intracellular imaging[42]. However, their potential in vitro applications remain largely unexplored. We propose that GECFINDER may serve as a valuable high-throughput screening tool for the detection of organic acids and their derivatives during the screening of high-yielding strains. FADS has been developed into an effective platform for high-throughput screening and microvolume single-cell analysis[43]. We intend to demonstrate that GECFINDERs are compatible with and effective for FADS.

We first demonstrated this application by developing a FADS-based high-throughput screening method to identify high phenylalanine-producing strains. Generally, industrial amino acid-producing strains produce phenylalanine at the grams per liter level, which greatly exceeds the dynamic range of GECFINDER-Phe3. Thus, we had to re-engineer the affinity of GECFINDER-Phe3 without affecting its ligand specificity, for phenylalanine to enable the screening of industrial phenylalanine-producing strains First, we performed virtual alanine scanning on the amino acid residues near the substrate pocket of GECFINDER-Phe3 using Discovery Studio 2019. The results showed that the mutation of F234A could reduce the affinity of GECFINDER-Phe3 for

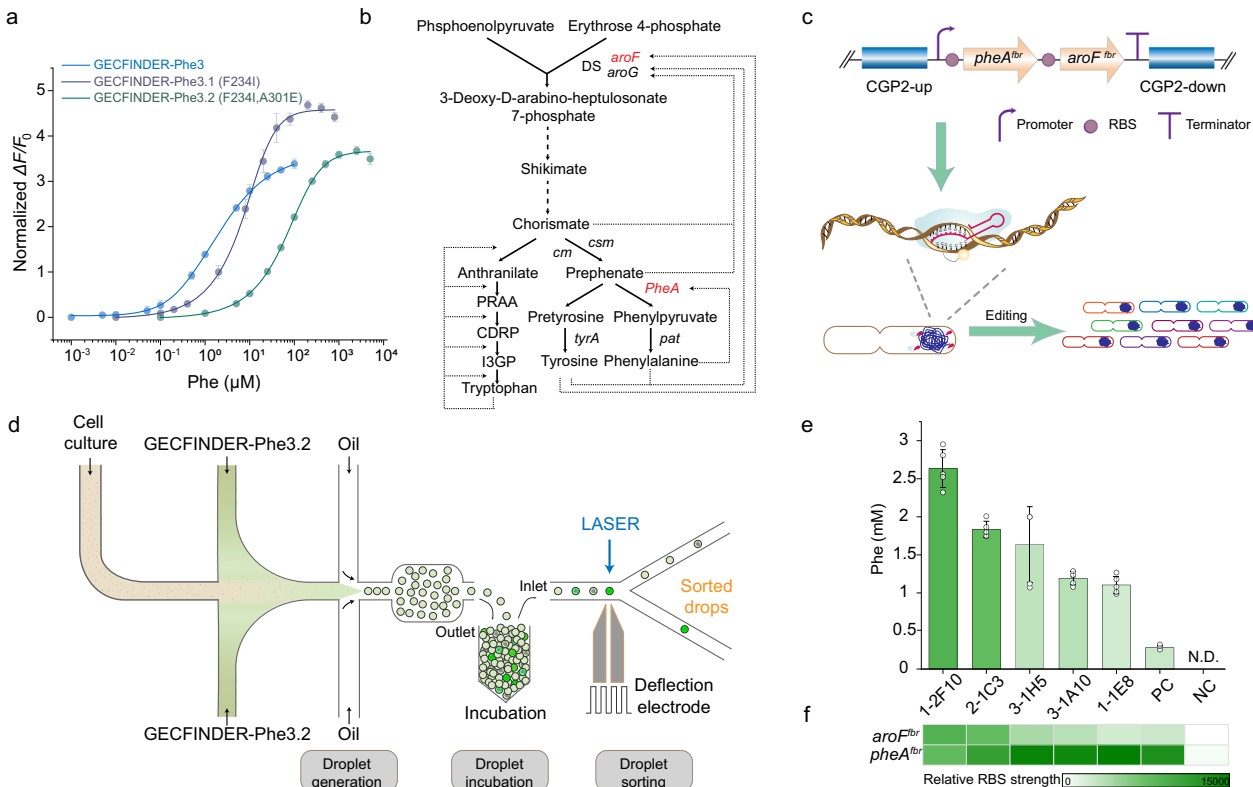

**Fig. 5 | GECFINDER applies for FADS sorting. a** Dose−response curves of purified GECFINDER-Phe3, 3.1 and 3.2 with increasing concentrations of phenylalanine. **b** Phenylalanine biosynthesis pathway in *C. glutamicum*. Dotted and dashed lines represented feedback inhibition and repression, respectively. Two target genes were highlighted in red. DS 3-deoxy-Darabinoheptulosonate 7-phosphate synthase, CM chorismate mutase, PDT prephenate dehydratase, CDRP 1-(2-carbox-yphenylamino)-1deoxy-D-nbulose-5-phosphate, I3GP indole 3-glycerolphosphate, PRAA n-(5-phospho-b-D-nbosylanthranilate), PRT anthranilate phosphoribosyl-transferase. **c** The artificial clusters equipped with constitutive promoters and tai-lored GGGGGGGG RBSs. Then BETTER used a CRISPR-guided ncas9-cytidine

deaminase fusion to generate G/A-rich RBS libraries from a custom initial RBS with eight consecutive Gs to generate variant cells with different phenotypes. **d** Droplet generation and sorting scheme combining GECFINDER and FADS. **e** Phenylalanine productions of *C. glutamicum* 1−2 F10, 2−1 C3, 3-1 H5, 3-1 A10, 1-1 E8, PC and NC in CGXII medium. ND represented "not detectable". Error bars reflected the mean ± S.D. of five biological replicates (*n* = 5). **f** RBS strength assay of the NC, PC strains and five screened strains with higher phenylalanine production. RBS strength was determined using a GFP reporter system. Scale bar represented GFP fluorescence normalized with OD600 nm. Values reflect the mean ± S.D. of three biological replicates (*n* = 3). Source data are provided as a Source data file.

phenylalanine (Supplementary Fig. 8a). Subsequently, we performed virtual saturation mutations on the F234 site (Supplementary Fig. 8b) and selected F234I, which has a medium mutation energy value ranking, for experimental verification. The experimental results showed that the F234I mutation increased the EC50 value of GECFINDER-Phe3 from 1.91 ± 0.08 to 7.51 ± 0.83 μM, reducing the affinity of GECFINDER-Phe3 for phenylalanine by almost 4-fold (Fig. 5a). To further reduce the affinity of the F234I mutant (GECFINDER-Phe3.1) for phenylalanine, a site-saturation library of A301 based on GECFINDER-Phe3.1 was created and screened. As a result, a mutant harboring a double mutation of F234I and A301E (GECFINDER-Phe3.2) with a EC50 of 67.63 ± 1.96 μM was obtained. The affinity of GECFINDER-Phe3.2 for phenylalanine was about 35-fold lower than that of GECFINDER-Phe3 (Fig. 5a), essentially satisfy-ing the requirements for the preliminary screening of high phenylalanine-producing strains. The substrate specificity and thermal stability of GECFINDER-Phe3.2 were evaluated. GECFINDER-Phe3.2 exhibited enhanced substrate specificity compared to GECFINDER-Phe3. GECFINDER-Phe3.2 demonstrated selective response to phenylalanine at higher substrate concentrations, without responding to leucine, methionine, tyrosine, and tryptophan, which induced response in GECFINDER-Phe3 (Fig. 4b and Supplementary Fig. 8c). Furthermore, the introduced point mutation in GECFINDER-Phe3.2 did not compromise its thermal stability, thus enabling stable operation and selective response to phenylalanine in the droplet microfluidic environment (Supplementary Table 1).

The strategies for regulating metabolism to increase phenylala-nine biosynthesis primarily focus on augmenting precursor substance accumulation, removing feedback inhibition of key enzymes, and modifying transport systems[44]. In this study, we propose utilizing the BETTER base editing method[45] to edit the ribosome binding site (RBS) in front of key enzymes within the phenylalanine biosynthetic path-way, generating an RBS mutation library (Fig. 5b, c). Then use GECFINDER-Phe3.2, in combination with FADS, as a high-throughput screening method to screen high-production strains of phenylalanine (Fig. 5d). This approach will enable us to explore the relationship between key enzymes expression intensity and phenylalanine yield. The rate-limiting enzymes 3-deoxy-D-arabinoheptulosonate-7-phos-phatesynthase (DAHPS, encoded by the *aroF* and *aroG* genes) and prephenate dehydratase (PDT, encoded by the *pheA* gene) are both feedback-inhibited by phenylalanine (Fig. 5b) in the phenylalanine biosynthesis pathway of *Corynebacterium glutamicum*[46,47]. The *pheA^fbr* and *aroF^fbr* genes from *Escherichia coli* MG1655[48,49], which were relieved from feedback repression, were integrated into the genome of *C. glutamicum* ATCC13032. The BETTER base editing method[45] was used to reprogram the expression of *pheA^fbr* and *aroF^fbr* to optimize the metabolic fluxes of the shikimic acid and branchial acid biosynthetic pathways for phenylalanine overproduction in *C. glutamicum* ATCC13032 (Fig. 5c). The artificial *pheA^fbr*-*aroF^fbr* cluster with tailored GGGGGGGG ribosome binding sites (RBSs) was inserted into the CGP2 prophage region of the *C. glutamicum* ATCC13032 chromosome. After

base editing, targeted next-generation sequencing revealed 5824 G/A/C/T-containing RBS types for *pheA^fbr* (containing 254 G/A types) and 4843 for *aroF^fbr* (containing 237 G/A types), indicating abundant RBS diversity in the screening library (Supplementary Fig. 8d–g).

To optimize the FADS screening parameters, the *pheA^fbr*-*aroF^fbr* cluster with GGGGGGGG or GAAAGGAG RBSs (commonly used as strong RBS) was inserted into the CGP2 prophage region of the *C. glutamicum* ATCC13032 chromosome to construct the Phe negative (NC) and Phe positive (PC) strains, respectively. The NC and PC strains were statically cultured at 30 °C in an incubator to simulate the growth state of cells in droplets. The phenylalanine production of PC after 16–24 h of static cultivation was 43.5–134.9 µM, within the detection range of GECFINDER-Phe3.2 (Fig. 5a and Supplementary Fig. 8i). The fluorescence of PC during continuous stationary incubation in droplets started to increase after 6 h, and its fluorescence difference from NC was most obvious at 12–24 h (Supplementary Fig. 8h). Therefore, the sorting time was set at 16 h. We constructed an artificial library by mixing NC and PC strains in a 99:1 ratio for the proof-of-concept screening. During sorting, the top 0.4% of droplets with the strongest green fluorescence signal were forced to deflect at a voltage of 500 V by FADS (Fig. 5d). Colony PCR was used to identify the NC and PC strains after the sorted droplets were spread. Among the surviving colonies, 16.7% were PC strains, showing an enrichment ratio[50] of 32 (Supplementary Fig. 8j), it means that GECFINDER-Phe3.2 can efficiently enrich strains with high phenylalanine production. This result demonstrated that GECFINDER-Phe3.2 combined with FADS can be used to effectively screen out the high phenylalanine-producing strains.

Then, we screened the *C. glutamicum* mutant library generated after the BETTER method using FADS. The library was co-encapsulated in droplets with GECFINDER-Phe3.2, and FADS analysis was performed after static cultivation for 12 h (Fig. 5d). Different ratios of the droplets with the strongest signals were sorted, collected, and then spread. We picked mutants from the agar plate into 96-well plates and fermented them for 24 h before screening their phenylalanine concentration using GECFINDER-Phe3 as a reagent. Finally, we selected five strains with higher phenylalanine production than PC and sequenced their RBSs (Supplementary Fig. 8k). To verify whether the improvement in phenylalanine production was solely due to the change in RBS strength, we rebuilt the selected strains from the wild-type strain and quantified their phenylalanine production after 24 h of fermentation in a 96-well plate by GECFINDER-Phe3. The reconstructed strains showed almost the same phenylalanine production capacity as the screened strains (Fig. 5e and Supplementary Fig. 8k), among which three strains generated yields more than 5-fold higher than that of PC. The strain 1–2 F10 accumulated the highest titer, i.e., 2.6 mM phenylalanine, nearly 10 times higher than that of the PC strain. Then the shake flask fermentation yield of 1–2 F10 was measured, resulting in a phenylalanine yield of 1.19 g/L after a 72-h fermentation period.

To characterize the RBS strength of the five phenylalanine-producing strains we obtained, the first 180 bp of the respective genes (*pheA^fbr*/*aroF^fbr*) with their RBSs and a green fluorescent protein gene were fused using a flexible linker (GGGGS)₃ to construct an RBS strength reporter system (Supplementary Fig. 8m). By measuring the RBS strength of *pheA^fbr* and *aroF^fbr*, we found that there was a clear positive correlation between phenylalanine production and the expression level of *aroF^fbr* (Fig. 5e, f). This result indicated that the DAHPS enzyme encoded by *aroF^fbr* is an important rate-limiting enzyme in the phenylalanine biosynthetic pathway and therefore upregulating the expression of DAHPS contributes to the enhancement of phenylalanine production. In conclusion, we developed a high-throughput screening method by combining GECFINDER with FADS and successfully screened strains with increased phenylalanine production. This method can be easily adapted for the high-throughput screening of other industrial amino acid- or other acid-producing strains.

## Discussion

Here we demonstrated that enzymes such as ANL superfamily enzymes can be converted into biosensors for various applications. We generated eight small-molecule GECFINDERs for detecting 4-coumaric acid, ATP, phenylalanine, glutamic acid, proline, isoleucine, histidine, and benzoic acid. In addition, by engineering the substrate-binding pocket of the LBD, we obtained three additional small-molecule GECFINDERs for leucine, tyrosine, and S-β-phenylalanine. To understand the working principle of GECFINDERs, we demonstrated that the catalytic capability of the first adenylation step was maintained, and the second thioesterification step was significantly impaired, which did not affect the accuracy of GECFINDER- mediated applications.

We also developed two approaches for the in vitro application of GECFINDER, which are generally underexplored in the field of cpFP-based biosensors. We first demonstrated that GECFINDERs are effective reagents for easy, rapid, and precise measurement of corresponding ligands in complex biological samples. In particular, we showed that GECFINDER-Phe3 can rapidly and accurately determine the phenylalanine concentration in human blood, indicating the potential for use as a POCT method for IEM. In 2018, Lin et al. suggested the use of biosensors based on the transcription factor to detect phenylalanine levels in urine[51]. This method involved co-culturing *E. coli* containing transcription factor-based biosensors with urine for a specific duration and subsequently assessing the concentration of phenylalanine by measuring fluorescence intensity. In contrast, GECFINDER offers a more straightforward and faster alternative, providing measurement results within just 5 min. We expect that the GECFINDERs generated in this work will be rapidly developed to track and diagnose other IEMs, such as homocystinuria, tyrosinemia and maple syrup urine disease, providing a simple and low-cost technique that will be especially useful in low-income regions. In addition, by combining GECFINDER-Phe3.2 with FADS, we rapidly screened out phenylalanine high-producing *C. glutamicum* strains with 10-fold higher yields than the artificially constructed high-yield strains from a complex library with millions of mutants generated using the BETTER method. Our results provide a standard example for the development of GECFINDERs to screen for other high-yielding strains. In the future, GECFINDER can be combined with FADS for the direct screening of high-yielding strains of economically valuable organic acids.

In this study, we expanded biosensor LBDs into the large reservoir of the ANL superfamily. The LBDs of GECFINDERs developed in this study were mainly based on the NRPS A domain subfamily of the ANL superfamily, and 544 small molecules (acids) have been identified as potential substrates of various A domains[52]. Theoretically, it is possible to rapidly construct genetically encodable biosensors for all these small molecules using the screening method developed here. In addition to the A domain of NRPS in the ANL superfamily, among more than 50,000 corresponding biosynthetic gene clusters from various forms of life, there are many aryl-CoA ligases, acetyl-CoA synthetases, fatty acyl-AMP ligases and fatty acyl-CoA synthetases that have been newly recruited as ANL superfamily members[53] and can be used as GECFINDER LBDs. Furthermore, we developed a benzoic acid biosensor that utilizes SrCAR as the LBD, which is distantly related to the ANL superfamily. These results expanded the scope of potential GECFINDER LBDs, and we expect to include more proteins with significant allosteric effects into the GECFINDER screening system in the future. In addition to the naturally occurring LBD from the ANL superfamily enzyme and similar enzymes, rational engineering and/or directed evolution can be easily applied to engineer LBDs that recognize unnatural ligands with great selectivity and tunable dynamic ranges for different applications.

Although we demonstrated that the LPXP motif is a suitable insertion site, it is still far from a sophisticated linker-designing rule.

Nevertheless, during the screening process, we collected a large amount of linker sequence data as well as their corresponding GECFINDER performance. In the future, we hope to exploit machine learning and the crystal structure of GECFINDERs for efficient de novo design of linkers to further shorten the GECFINDER construction time.

The intracellular application of fluorescent biosensors offers exciting possibilities for studying metabolic dynamics within living cells. While our current study focuses on characterizing and evaluating GEC-FINDER as purified reagents, the potential for intracellular use remains an area of future exploration. One potential key challenge in the intracellular application of GECFINDER is its ATP dependency, which can potentially impact the cellular environment and limit its utility. In general, intracellular ATP concentrations in animal and microbial cells range from 2.74 to 7.47 mM[54], providing sufficient ATP for GECFINDER. Moreover, due to the insertion of cpEGFP, GECFINDER only undergoes a partial reaction, and the labile intermediate cannot be effectively released, resulting in a significantly slower ATP consumption process. This extremely slow ATP consumption rate is unlikely to hinder the intracellular application of GECFINDER. Furthermore, we can further weaken or eliminate GECFINDER's dependence on ATP through protein engineering. Additionally, achieving a high signal-to-noise ratio poses another challenge for intracellular application. Although some GEC-FINDER variants (Ile, His, Pro) currently exhibit lower signal-to-noise ratios, we have shown that protein engineering can rapidly improve the performance of various GECFINDERs. These advancements will contribute to accurate measurements of intracellular metabolic dynamics, enhancing the utility of GECFINDER in biological applications. Altogether, we expect GECFINDERs to provide an unprecedented toolbox for small-molecule acid detection in terms of number and applicability.

Here we have illustrated that enzymes can be easily converted to biosensors, representing a largely untapped resources for biosensor design and creation. In the development of cpEGFP-based biosensors, the selection of the appropriate LBD for substrate coupling conformational change is followed by the selection of an appropriate cpEGFP insertion site and optimization of the cpEGFP linker to enhance biosensor performance. Among these steps, selecting a suitable cpEGFP insertion site is the most critical for cpEGFP-based biosensor construction. Without the aid of crystal structure to clarify conformational changes, finding a suitable cpEGFP insertion site can be challenging, necessitating the construction of numerous mutants. However, with GECFINDER, homology modeling can be used to locate the positions of Hinge A and B, thus limiting the insertion site of cpEGFP to a narrow range, simplifying the construction of cpEGFP-based biosensors. GEC-FINDER takes advantage of the ligand-binding-induced domain motion of ANL superfamily enzymes and carboxylic acid reductase and inserts cpEGFP into the flexible hinge between domains to generate small-molecule biosensors. We expect that the biosensor construction method developed in this study will be directly applied to other enzymes with large-scale ligand-coupled domain movements to generate biosensors. In the Protein Structural Change Database[55], there are nearly 50 representative enzymes with hinge-mediated ligand-coupled domain motions, all of which are highly promising targets to be converted into biosensors. Moreover, a broad range of enzymes with allosteric effects[56] may also serve as biosensor LBDs when properly fused with cpFP. In particular, some enzymes that follow the open–close mechanism, such as kinases[57], transferases[58], oxidoreductases[59], and lyases[60], have the great potential to be coupled with cpFP to generate cpFP-based biosensors for a variety of compounds.

## Methods
### Ethical statement
This study involves human participants and was approved by the Ethics Committee of Tianjin Institute of Industrial Biotechnology, Chinese Academy of Sciences (TIB202306-002). Informed written consent was obtained from all human blood sample donors, and their participation in the study was voluntary.

### Strains and culture conditions
Strains used in this study are listed in Supplementary Data 3. *E. coli* DH5α was used for plasmid construction and was cultivated in LB medium at 37 °C. *E. coli* BL21 (DE3) or BAPI was used for protein expression in LB medium at 37 °C or 16 °C, supplemented with 0.4 mM of isopropyl β-d-thiogalactoside (IPTG) as inducer when needed. *C. glutamicum* ATCC 13032 was cultured in LBHIS or CGX II medium at 30 °C or 37 °C. When necessary, kanamycin (Km) was added in the final concentration of 25 μg/mL.

### General plasmid construction
Plasmids used in this study are listed in Supplementary Data 3. Primers used for cloning are listed in Supplementary Data 4. The corresponding DNA fragments were amplified by PCR (Q5 high-fidelity DNA polymerase, NEB), and the plasmids were constructed by homologous recombination using the ClonExpress® II One Step Cloning Kit (Vazyme, Nanjing, China) according to the manufacture protocols.

### Construction of GECFINDER mutant libraries
The sources of all utilized LBDs are listed in Supplementary Data 4. To construct the GECFINDER mutant library, the cpEGFP with random two-amino-acid linkers at both ends was integrated into hinge A or hinge B of LBD. Initially, the LBD fragment was amplified via PCR from the template (Supplementary Data 4), and subsequently, the purified PCR product was cloned into pET28a (+) by homologous recombination method to generate the pET28a_LBD plasmid. Once the insertion site was confirmed, PCR was conducted with pET28a_LBD as the template, and the primer pair was segmented into three parts: 20-bp cpEGFP homology arm, 6-bp degenerate sequences, and 20-bp binding region. The purified fragment was merged with cpEGFP by homologous recombination. After ligation, the product was digested with *Dpn* I enzyme and then transformed into *E. coli* BAPI.

### High-throughput screening of effective GECFINDERs from mutant libraries
Kanamycin-resistant *E. coli* BAPI colonies were picked and grown overnight in 96-well plates with LB medium and kanamycin at 37 °C overnight. The resulting seed culture was then used to inoculated another set of LB-containing 96-well plates and cultivation at 37 °C until the cell density ($OD_{600}$) reached 0.6–0.8. Subsequently, 0.4 mM IPTG was added, and the culture was cultivated for an additional 48 h at 16 °C[61]. The cells were then collected by centrifugation, washed twice with PBS, and resuspended in PBS, followed by repeated freezing and thawing to release the intracellular protein. After centrifugation, the supernatant was transferred into microtiter plates, and $MgCl_2$ and ATP were added to a obtain final concentration of 2.5 mM and 1 mM, respectively. The fluorescence values were determined using a Microplate Reader (Synergy H4, Bio Tek) with excitation/emission wavelengths of 400 nm/ 510 nm and 480 nm/510 nm ($F_0$). Next, the small molecule effector (1 mM) was added to the microtiter plates, and fluorescence intensity was continuously measured at 400 nm/510 nm and 480 nm/510 nm for 15 min at one-minute intervals. The last reading at 480 nm/510 nm was recorded as fluorescence reading $F$ after measurement. Promising mutants were selected based on colonies with $(F - F_0)/F_0$ values beyond 0.1 or below −0.3 and subjected to re-screening and Sanger sequencing. Mutants with good reproducibility in the secondary screening were selected for purification and function verification.

### Protein expression and purification
*E. coli* BAPI was cultured overnight in kanamycin-containing LB medium at 37 °C and 220 rpm. The culture was then inoculated to fresh

kanamycin-containing LB medium and further cultivated at 37 °C until $OD_{600}$ reached 0.6–0.8. Induction was performed by adding 0.4 mM of IPTG and culturing for another 48 h at 16 °C.After harvesting the cells by centrifugation, they were resuspended in a buffer containing 20 mM Tris and 500 mM NaCl (pH = 7.9) and disrupted by sonication. The supernatant obtained after centrifugation was loaded into a 5 mL His-Trap column (GE Healthcare) on fast protein liquid chromatography (AKTA purifier 10, GE Healthcare), which was washed with the washing buffer containing 20 mM Tris, 500 mM NaCl, and 20 mM imidazole (pH = 7.9). Protein was eluted from the column using an elution buffer with an increasing gradient concentration of imidazole (20–500 mM) in 20 mM Tris and 500 mM NaCl (pH = 7.9). Imidazole was removed by centrifugation using an ultrafiltration tube (30 kDa MWCO, Millipore) and purified protein was stored at −80 °C in individual aliquots in a buffer containing 20 mM Tris, 500 mM NaCl, and 10% glycerin (pH = 7.5).

## Determination of the dose–response curve, substrate specificity, fluorescent spectrum, pH stability and thermal stability

The study conducted dose–response curve, substrate specificity, and fluorescent spectrum analyses in 100-mM Tris buffer (pH = 7.5), while pH stability experiments were performed in 100-mM Tris buffer at three different pH values (6.5, 7.5, and 8.5) at room temperature. GECFINDER protein at a concentration of 0.75 μM, various small molecule concentrations, 1 mM ATP, and 2.5 mM $MgCl_2$ were used in the experiments. Fluorescent signals were detected using a Microplate Reader (Synergy H4, Bio Tek) at an excitation/emission wavelength of 460 nm/510 nm or 480 nm/510 nm. The dose-response curve was determined immediately after the addition of the analyte. The total duration of the measurement was 15 min, with readings taken at one-minute intervals. The dose-response curve was plotted using the fluorescence values obtained after achieving stability in the fluorescence signal. The dose-response curves were fitted using the logistic model LOGISTIC5 available in Origin 2021, which could provide a more adequate fit to asymmetric dose-response data[62,63]. Excitation spectra were scanned from 300 to 540 nm with a step size of 5 nm, and the emission wavelength was set to 560 nm. Emission spectra were scanned from 470 to 700 nm with a step size of 5 nm, and the excitation wavelength was set to 450 nm. The thermal stability of GECFINDER was analyzed using the UNcle multifunctional protein stability analysis system. Measurements were performed in 20 mM MOPS buffer with 2 mM DTT at a sample concentration of 1 mg/mL. Absolute fluorescence quantum yields were measured using an absolute photoluminescence quantum yield spectrometer (FS5, Edinburgh).

## LC-MS analysis of the Nt4CL2 and GECFINDER-4CA products

Reactions were carried out overnight at room temperature using Nt4Cl2 and GECFINDER-4CA at the following concentrations: 100 mM Tris (pH = 7.5), 2.5 mM $MgCl_2$, 2.5 mM ATP, 0.2 mM CoA (required for thioester product determination), 0.2 mM 4-coumaric acid, and 0.2 μM of enzyme. 10 μL of each reaction mixture was analyzed by negative-ESI LC/MS using an Agilent 1200 series HPLC system equipped with a dC18 column (Atlantis T3 5 μm 4.6 × 250 mm, Waters) coupled to a Bruker microTOF-Q II mass spectrometer. The mobile phases were water (buffer A) and acetonitrile (buffer B) and a linear gradient of 5–95% B was applied at a flow rate of 0.3 mL/min over 25 min. The 4-coumaroyl-AMP and 4-coumaroyl-CoA were detected at 340 nm and 333 nm, respectively[25]. UNIFI (v 1.9.4.053, Waters, Corp.) for Q-TOF MS (Waters, Wilmslow, UK) was used for collection of LC-MS data.

## Detection of GECFINDER AMP generation by HPLC

All reactions were conducted at room temperature in the buffer containing 50 mM Tris, 300 mM NaCl, 10 mM $MgCl_2$ (pH = 8). The final concentrations of proteins (GrsA-PheA Apo, GrsA-PheA holo,

GECFINDER-Phe3 and GECFINDER-Phe3ΔPCP) and ATP were 7 μM and 1 mM, respectively[37]. To initiate the reaction, 1 mM phenylalanine was added, and the reaction was stopped after 15 min by ultrafiltration. The control reaction was carried out under the same conditions as described above, except that the proteins were absent. Liquid chromatography was performed using an HPLC system (1260, Agilent) with a dC18 column (Atlantis T3 5 μm 4.6 × 250 mm, Waters), monitored at 254 nm. The mobile phase consisted of 2.5% methanol and 25 mM potassium phosphate buffer (adjusted to pH = 6.5 with $H_3PO_4$) for phase A, and 50% methanol and 25-mM potassium phosphate buffer (adjusted to pH = 6.5 with H3PO4) for phase B[64]. The gradient elution was set as follows: 0–2 min, 100% A; 2–25 min, 100% A to 33 % A; 25–30 min, 33% A and 67% B, with a flow rate of 0.8 mL/min[37]. OpenLAB CDS ChemStation Edition for Agilent 1260 Infinity II HPLC system (Agilent, Waldbronn, Germany) was used for collection of HPLC data.

## ESI-TOF-MS assay

The purified proteins were desalted using ultrafiltration tubes (30 kDa MWCO, Millipore) before MS analysis. GrsA-PheA holo or GECFINDER-Phe3 at a concentration of 6 μM were mixed with 1 mM ATP and 1 mM phenylalanine at room temperature for 12 h before MS analysis. The GrsA-PheA holo or GECFINDER-Phe3 protein was desalted using a buffer containing 70% water, 30% acetonitrile and 0.1% formic acid. MS experiments were conducted in the positive ion mode using a TripleTOF 6600. The mobile phase used was 50% acetonitrile, 50% water, and 0.1% formic acid. The ESI-TOF parameters were set according to Sun et al.[37], specifically, the gas temperature was 325 °C, Vcap was 350 V, the nebulizer pressure was set at 30 psig, the fragmentor voltage was 175 V, and the skimmer voltage was 65 V. The TOF mass window was set to be in the range of 100 to 1600 $m/z$. Analyst TF (v 1.8.1) for TripleTOF 6600 (SCIEX, MA, USA) was used for collection of MS data.

## Detection of amino acid concentration in biological samples using GECFINDERs

Fingertip blood was obtained from three volunteers, and the serum samples were collected after centrifugation. Glutamic acid fermentation samples and glutamic acid concentration results from SBA were provided by Ping Zheng's laboratory. All reactions were performed in the 100 mM Tris buffer (pH = 7.5) containing 1 mM ATP, 2.5 mM $MgCl_2$, 3 μM GECFINDERs. Standard amino acid solutions of various concentrations (phenylalanine: 0.5–5 μM, glutamic acid:10–1000 mg/L) or biological samples were added to the buffer solution (5 μL of serum added to a 200 μL system, 10 μL of fivefold diluted fermentation broth added to a 200 μL system). The mixture was incubated at room temperature for 5 min, and fluorescence intensity was measured using a fluorescence microplate reader at the excitation/emission wavelengths of 460 nm/510 nm. The standard curve was drawn using the concentrations of the standard amino acid solutions as the abscissas and the fluorescence intensities of GECFINDER as the ordinates. Linear fit of the standard curve and sample concentration calculation were performed using Origin 2021 software. Sample concentrations were calculated based on the standard curve. The information about the gender and age of the research participants are listed in Supplementary Table 3.

## Detection of phenylalanine concentration in serum using HPLC

Standard phenylalanine solutions were prepared using 2.5% perchloric acid at concentrations of 10 μM, 37.5 μM, 75 μM, and 100 μM. After mixing the serum sample with an equal amount of 5% perchloric acid by vortexing for 20 s, the mixture was kept at room temperature for 10 min. The supernatant was then collected by centrifugation and filtered through a 0.2-μm nylon filter. The filtered samples were kept on ice before HPLC analysis. HPLC experiments were carried out using an

Agilent 1260 HPLC system equipped with a reversed-phase column dC18 (Atlantis T3 5 μm 4.6 × 250 mm, Waters) and the signals were monitored at 190 nm. The mobile phases consisted of 90% ultra-pure water (buffer A) and 10% acetonitrile (buffer B). The flow rate was set to 0.5 mL/min and the column temperature was maintained at 25 °C. The standard curve was generated using the standard phenylalanine solutions, with their concentrations on the x-axis and peak areas on the y-axis. Linear fit of the standard curve and sample concentration calculation were performed using Origin 2021 software. The concentration of phenylalanine in serum samples was calculated based on this standard curve.

## Chromosomal integration

To optimize the droplet microfluidic screening method, the pK18mobsacB plasmid with corresponding transcriptional translation elements and 1000 bp homologous arms at both ends was used to integrate the $pheA^{fbr}$ and $aroF^{fbr}$ genes with RBS "GGGGGGGGG" or "GAAAGGAG" into the genome of *C. glutamicum* ATCC 13032, and the phenylalanine low (Negative Control, NC) and high (Positive Control, PC) yield controls were constructed. Plasmids pK18mobsacB_NC and pK18mobsacB_PC were electroporated into *C. glutamicum* ATCC 13032. The resultant cell suspensions were diluted by 10-fold with BHI, regenerated at 30 °C and 220 rpm for 2 h, and incubated on the LBHIS plates (supplemented with 25 μg/mL of kanamycin) at 30 °C. The emerging colonies were picked and subjected to colony PCR to verify the successful homologous recombination on the genome. The correct single-exchange strains were picked and incubated in LBHIS medium overnight at 30 °C, 220 rpm. Then the culture was appropriately diluted, spread on LAS plates and incubated at 30 °C. Subsequently, the colonies on LAS plates were picked and cultured simultaneously on the LBHIS plates with and without 25 μg/mL of kanamycin at 30 °C. The colonies that could not survive on the kanamycin-containing plates were selected for colony PCR validation and sequencing.

## Droplet microfluidics proof-of-concept screening

PC and NC strains were inoculated into CGX II medium with 20 g/L glucose and cultivated for 16 h at 30 °C. The seed cultures were transferred into fresh CGX II medium with 20 g/L glucose and further cultivated for 10 h at 30 °C. The NC and PC bacterial cells were collected by centrifugation at 1700 × g for 2 min, washed three times with CGXII containing 60 g/L of glucose, and resuspended in CGXII containing 60 g/L of glucose at an OD$_{600}$ of 0.2. An artificial library was generated by mixing NC and PC suspensions at a ratio of 99:1. In addition, GECFINDER-Phe3.2, ATP and MgCl$_2$ were added to CGX II with 60 of g/L glucose at final concentrations of 12 μM, 2 mM and 5 mM, respectively. Droplets were then generated from NC strain, PC strain and artificial library at rates of 400 μL/h for the oil channel, 100 μL/h for the bacterium channel, and 100 μL/h for the sensor channel, producing droplets with diameters of 20–25 μm. The droplets were incubated at 30 °C for 6, 12, 24, and 36 h for droplet observation and sorted at 16 h. The top 0.4% of droplets with the highest green fluorescence signal were forced to deflect in the voltage of 500 V by FADS. The screened droplets were spread on LBHIS plates and incubated for 36 h at 30 °C. Finally, the ratio of PC and NC strains was determined by colony PCR.

## Generation of RBS variants with base editing

The NC strain was co-transformed with PXMJ19-Cas9-AID-TS and PTrcmob-rbs2-2gRNA$^{no\ per}$ plasmids. The resulting transformants were grown overnight in LBHIS medium supplemented with 5 μg/mL of chloramphenicol and 15 μg/mL of kanamycin followed. The induction editing was performed in CGX II supplemented with 5 μg/mL of chloramphenicol, 15 μg/mL of kanamycin, and 18 g/L of glucose at an initial OD$_{600}$ of 0.2 and IPTG at a concentration of 0.4 mM for 20 h at

30 °C[45]. the plasmid was then eliminated by incubation the culture overnight in LBHIS at 37 °C.

## Next-generation sequencing (NGS) and data analysis

Three biological replicates of cell cultures after base editing were mixed in equal proportions and used for genomic DNA extraction. DNA fragments, spanning 150–200 bp, were amplified by PCR using the primers specified in Supplementary Data 4 from the extracted genomic DNA. Library construction and sequencing were performed by Novogene (Beijing, China) and approximately 100,000 reads were analyzed per sample. llumina Hiseq Control software was used on the Illumina Hiseq sequencers to collect the sequencing data. For NGS data analysis, 20 nT sequences upstream of the target region were used to locate the position of the target region in each read by BLAST. The sequence of the target region was then extracted from the reads and mapped to a reference sequence to analyze base editing events. Each RBS variant was quantified using the R package (v4.0.2). Sequence logos were generated using the statistics of the RBS variants containing G/A/C/T and the R package "ggseqloo" (v0.1). Heat maps were generated using the statistics of RBS variants containing G/A and the R package "ggplot2" (v3.3.2).

## High-throughput screening for phenylalanine high-producing strains by FADS

After editing and plasmid curing, the RBS library was cultured in CGX II medium with 20 g/L glucose for 10–12 h at 30 °C. The bacterial cells were collected by centrifugation, washed thrice with fresh CGX II medium, and resuspended in CGX II with 60 g/L of glucose. The suspension was adjusted to an OD$_{600}$ of 0.2. GECFINDER-Phe3.2, ATP and MgCl$_2$ were added to CGX II (60 g/L glucose) at final concentrations of 12 μM, 2 mM and 5 mM, respectively. Droplets were generated on the chip with rates of 400 μL/h for the oil channel, 100 μL/h for the bacterium channel, and 100 μL/h for the sensor channel, resulting in droplets of 20–25 μM in diameter. The droplets were incubated in a syringe for 10–12 h at 30 °C, and then sorted. The droplets with the strongest signal were forced to deflect in 500 V of voltage by FADS, with varying ratios. The sorted droplets were plated on LBHIS plates and incubated at 30 °C for 36 h. The colonies were then picked into 96-well plates containing CGXII and cultured overnight at 30 °C and 800 rpm. the culture was transferred to another 96-well plate with CGX II containing s 60 g/L glucose and fermented for 24 h at 30 °C, 800 rpm. After fermentation, the supernatant was separated from the bacterial cells by centrifugation at 3000 × g for 10 min. The supernatant was mixed with GECFINDER-Phe3 (2 μM), ATP (1 mM) and MgCl$_2$ (2.5 mM) in 100 mM Tris buffer (pH = 7.5) and allowed to stand for 30 min. Then, the fluorescence of the mixture was measured at excitation/emission wavelengths of 460 nm/510 nm. RBS variants with high fluorescence intensity were selected for sequencing.

## Shake flask fermentation of 1–2 F10

The bacterial strain 1–2 F10 was activated by inoculating on solid BHI or LBHIS plates. Single colonies were picked and transferred to seed culture medium (CGXII + 20 g/L glucose) in 250 mL shake flasks and incubated overnight at 30 °C. For fermentation, 10% of the seed culture was inoculated into the fermentation medium (CGXII + 60 g/L glucose) in 500 ml shake flasks and incubated at 30 °C for 12–84 h. The fermentation broth was collected and the phenylalanine yield was measured with GECFINDER-Phe3. Three biological replicates were performed.

## Detection of phenylalanine concentration in fermentation broth using GECFINDER-Phe3

All reactions were carried out in 100 mM Tris buffer (pH = 7.5) containing 1 mM ATP, 2.5 mM MgCl$_2$, 0.2 μM GECFINDER-Phe3, and standard phenylalanine solutions of certain concentrations of 0.1, 0.25, 0.5,1, 2.5, and 5 μM as well as fermentation broth samples, were used.

The fermentation broth samples were dilute to an appropriate concentration (50–100 times). The buffer, ATP and MgCl$_2$ were pre-mixed and dispensed into a 96-well plate, followed by addition of standard phenylalanine solutions or fermentation broth samples. After incubation at room temperature for 5 min, fluorescence intensity was measured at the excitation/emission wavelengths of 460 nm/510 nm using a fluorescence microplate reader. The standard curve was plotted using the concentrations of standard phenylalanine solutions as the abscissas and the fluorescence intensities of GECFINDER-Phe3 as the ordinates. Linear fit of the standard curve and sample concentration calculation were performed using Origin 2021 software. GECFINDER fermentation broth concentrations were calculated using standard curve.

### Determination of RBS strengths

The pEC-XK99E plasmids with the RBS strength reporter system (listed in Supplementary Data 3) were electroporated into *C. glutamicum* ATCC 13032 and spread on LBHIS agar plates supplemented with kanamycin (25 µg/mL). After static cultivation at 30 °C for 36 h, the resulting colonies were transferred to a 96-well plate containing LBHIS with kanamycin (25 µg/mL) and cultured at 30 °C and 800 rpm for 16 h. The bacterial broth was then transferred to a 96-well plate containing fresh LBHIS with kanamycin (25 µg/mL) and cultured at 30 °C and 800 rpm until reaching the stationary growth phase. The cells were harvested by centrifugation at 2000 × *g* for 15 min, washed three times with PBS, and resuspended in PBS buffer. The OD$_{600}$ and GFP fluorescence signal (excitation wavelength = 488 nm, emission wavelength = 520 nm) of the bacterial suspension were measured by the microplate reader (Synergy H4, Bio Tek).

### Statistical and reproducibility

All experiments were performed for at least two independent biological replicates. No statistical method was used to predetermine sample size. No data were excluded from the analyses. The experiments were not randomized. The Investigators were not blinded to allocation during experiments and outcome assessment. Two samples were compared using Student's two-tailed *t* test.

### Reporting summary

Further information on research design is available in the Nature Portfolio Reporting Summary linked to this article.

## Data availability

The data supporting the findings of this work are available within the paper and the Supplementary Information files. The protein structure data used this study are available in the RCSB PDB under accession codes 1AMU, 5U95, 5BSM, and 5BST. The raw reads of the NGS data were deposited into the Sequence Read Archive database of NCBI under accession number PRJNA1019141. Source data are provided with this paper.

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

## Acknowledgements

We thank Jian Zhou Chen from the Ping Zheng's Laboratory for providing us with glutamic acid fermentation samples and SBA measurement results. This work was supported by the National Key R&D Program of China (2021YFC2103300 to M.W.), the National Natural Science Foundation of China (32001050 to W.P., 32200043 to Y.Z. and 32101186 to Y.M.), "Key Research Project" of Haihe Laboratory of Synthetic Biology (22HHSWSS00030 to M.W.), the Youth Innovation Promotion Association of the Chinese Academy of Sciences (2023187 to Y.Z.), and the Tianjin Synthetic Biotechnology Innovation Capacity Improvement Project (TSBICIP-PTJS-003 to M.W.).

## Author contributions

M.W., J.W. and R.T. conceived and designed this project. J.W., N.X., W.P. and S.L. performed the experiments. J.W., N.X., X.N. and W.Y. analyzed the data. J.W. wrote the initial paper draft. M.W., Y.Z., Y.M., Y.L., H.C. and Y.G. gave conceptual advice and modified the manuscript. M.W. finalized the manuscript.

## Competing interests

The authors declare no competing interests.
