## [Peer Review File · Nature Communications]

REVIEWER COMMENTS

Reviewer #1 (Remarks to the Author):

The article by Wang, J. et al entitled, “Converting biocatalysts into biosensors: repurposing conformational changes in ANL superfamily enzymes to detect various acids” describes the screening and generation of a series of fluorescent biosensors comprised of an enzyme and a circularly permuted GFP. The authors then use these tools to measure phenylalanine and glutamic acid in point-of-care diagnostics as well as in a high-throughput screening method for industrially-relevant yeast strains.

The authors extensively explore a family of enzyme and develop several fluorescent biosensors for acids, a couple of which show moderate fluorescence contrast. In general, the data support the conclusions of the authors. The article however suffers from a convoluted description of the protein engineering, some missing data, and a lack of perspective in the discussion.

The ANL family is quite large and comprises two major subclasses of enzyme: NRPS adenylation domains and acyl synthetases. The paper describes trying to generally engineer sensors in this family, although the most successful sensors all derive from the peptidyl carrier protein domain-containing NRPS adenylation domains. If the journal is amenable, it might help the clarity of the paper to move the description of some of the less successful engineering efforts to a supplemental note and focus the main text on protein engineering to describe the efforts that eventually resulted in the sensors used in the final applications.

There are several aspects of the data that are missing or need clarification. Specifically, the authors do not report the quantum yield or the brightness of the sensors, which has direct implications for how easy the sensors are to use. Additionally, since the sensors appear to complete half of their catalytic reaction, care should be taken in terms of using KD to describe their dose-response profiles to their ligands since the catalytic activity means that this is not a simple binding affinity. Did the dose-response profiles change over time? How long were the titrations incubated prior to measurement? Lastly, reported spectra of the sensors and their description require further clarification. Table 1 reports the excitation maximum of Phe3 to be 460 nm, yet a visual inspection of the spectra doesn't support this. Furthermore, some sensors appear to be ratiometric in excitation (GECFINDER-His2, 4CA), but are described in a confusing fashion.

The discussion is lacking in perspective that provides the reader with context to evaluate the sensors and their potential impact. For example, all of the demonstrations of the use of these sensor use them as purified reagents at relatively high concentrations. Yet, the authors suggest that in the future these sensors might be used to measure metabolites intracellularly. Not only were the sensors not

demonstrated to be compatible with in cellulo applications, but a sensor for metabolites that consumes ATP seems disadvantageous for capturing metabolic dynamics. It would also be helpful to contextualize the use of a fluorescent protein, rather than a system that might allow for amplification or provide more photons.

Some small changes:

The acronym POTS needs to be defined in the text

cpEGFP changed to cpFP in the middle of the text and CpFP in the legend of Figure 2. Please regularize.

Reviewer #2 (Remarks to the Author):

Wang et al. apply the strategy of fusing circularly permuted GFP to proteins that undergo a ligand-induced conformational changes to the ANL superfamily of enzymes. The novelty and impact of the present manuscript is in the application of this strategy to enzymes and the demonstration of its broad applicability. Several biosensors could be generated using rational design and screening methods and applications in sensing metabolites in blood or in screening for overproducing strains for biotechnological applications is demonstrated in extensive work.

The manuscript is generally sound and well written, however there are a number of grammatical problems or typos, however mostly easy to understand and correct. Also the figures are of high quality.

Major points

When trying to generate a GLU-sensor by inserting cpGFP into PpsA-GluA, the authors state that they generated a sensor that detected ATP, but not glutamate (Extended Fig. 1 a-c). Extended Fig. 1c indicates that this sensor would also detect Mg²⁺ ions. However, more importantly, the authors add Mg²⁺ and ATP to all assays. ATP is cosubstrate for the enzymatic reaction and it probably binds together with ATP. The authors do not describe to what extent the other sensors depend on the ATP and Mg²⁺ concentrations. This is important information that should be presented. While this may not be so relevant for the in vitro applications described by the authors, it would be relevant for potential in vivo applications.

The authors provide no information as to how the dose-response curves were fitted to obtain the specified KD values. The authors need to specify the exact mathematical formula used for fitting and specify for each curve which parameters have been obtained. Also the software should be stated. Some

dose-response curves appear as if they may be fitted by a simple sigmoidal curve without apparent cooperativity, such as Fig. 3a. In others the fluorescence changes take place over a wider concentration range, indicating that a Hill coefficient was probably used for fitting. Fig. 3a is a dose-response curve that likely does not support K_d fitting and this fluorescence change may be generated by other mechanisms than the ligand-induced conformational change. In curves extended Fig. 2 e and i as well as extended Fig. 3 a and g the fit curves quickly go into saturation (become horizontal) at higher substrate concentration and I really wonder which equilibrium binding model generates such a binding curve.

Minor points

Concerning the ANL enzymes used to design the biosensors. I would suggest to avoid the term biocatalyst and only use the term enzyme instead. "Biocatalyst" is usually (admittedly not always) used when referring to enzymes in a biotechnological sense for synthesis etc.

On page 4, line 73 the authors cite two papers to state that cpFP-based biosensors usually have wider dynamic ranges as FRET-based biosensors. Please check if the cited papers really describe a comparison of FRET-based biosensors and cpFP and if you want to confirm this statement. For a non-expert, one should better specify "...usually have wider dynamic ranges (concerning the fluorescence change) as..." as one might apply "dynamic range" also to the analyte concentration.

Page 4, line 80: "Catalytic enzymes..." Enzymes are always catalytic.

Fig.1 is a nice overview figure. I wonder if the authors could replace the figure of the instrument in "GECFINDER screening" by a scheme that illustrates the principle of the screening process.

Page 12, line 223: "By comparing the sequences of the GrsA-PheA with a known structure (PDB ID: 223 1AMU) and the other 160 A domains, the...": Replace "and the other" by "with".

In Table 2, "ND" is specified for the operating range and K_D for GECFINDER-Leu1 and Ligand Phe. Explain if ND stands for not determined or not detectable and why this is the case, although $\Delta F/F_0$ is reported.

Reviewer #3 (Remarks to the Author):

The manuscript from Wang et al., shows the repurposing of a common biosynthetic enzyme fold for a new utility as a biosensor. Specifically, the authors focus on enzymes of the ANL superfamily, which are known to undergo conformational rotation during the catalytic cycle. Insertion of circularly permuted GFP superfolder into the loop region between the adenylation N domain and the C domain or between the adenylation domain and carrier domain yields an enzyme that undergoes fluorescence shift in response to a substrate of choice. The idea is clever and may have some utility. While the current response requires low to high micromolar concentrations of substrate, this is something that can be improved upon in future iterations. Overall I am enthusiastic about this work but there are several points that need to be clarified in a revised version. Specifically,

1. I am confused why the repurposed catalysts do not turn over. The dose responses only show linear increases. The authors should clarify or provide a rationale for why this is so.

2. Figure 2 is entirely too confusing as drawn. Not all ANL enzymes have a PCP domain so the top half of the panels do not make sense in the context of panel A.

3. There should be a more robust discussion of signal/noise of the response, especially if the authors are going to propose this as a system for use in biology.

4. As I understand it, all of these engineered ANL enzymes require a concentration of ATP to maintain saturation. This is only briefly mentioned in the manuscript and should be stated more explicitly. Moreover, where possible, the known literature K_d values for ATP for the various systems under study should also be stated.

Reviewers' comments & the authors' replies

We gratefully thank the editor and all reviewers for their time spend making their constructive remarks and useful suggestions, which has significantly raised the quality of the manuscript and has enable us to improve the manuscript. Each suggested revision and comment, brought forward by the reviewers was considered and addressed accordingly. Below the comments of the reviewers are point by point response and the revisions are indicated.

Reviewer #1:

The article by Wang, J. et al entitled, “Converting biocatalysts into biosensors: repurposing conformational changes in ANL superfamily enzymes to detect various acids” describes the screening and generation of a series of fluorescent biosensors comprised of an enzyme and a circularly permuted GFP. The authors then use these tools to measure phenylalanine and glutamic acid in point-of-care diagnostics as well as in a high-throughput screening method for industrially-relevant yeast strains.

The authors extensively explore a family of enzyme and develop several fluorescent biosensors for acids, a couple of which show moderate fluorescence contrast. In general, the data support the conclusions of the authors.

Reply: We sincerely appreciate the reviewer's positive comments.

The article however suffers from a convoluted description of the protein engineering, some missing data, and a lack of perspective in the discussion. The ANL family is quite large and comprises two major subclasses of enzyme: NRPS adenylation domains and acyl synthetases. The paper describes trying to generally engineer sensors in this family, although the most successful sensors all derive from the peptidyl carrier protein domain-containing NRPS adenylation domains. If the journal is amenable, it might help the clarity of the paper to move the description of some of the less successful engineering efforts to a supplemental note and focus the main text on protein engineering to describe the efforts that eventually resulted in the sensors used in the final applications.

Reply: We gratefully appreciate your valuable suggestion. We agree that the protein engineering section could be clarified to enhance readability. In the revised version, we have moved the description of less successful engineering efforts, including the expansion of GECFINDER ligand binding domain and the preliminary efforts in rational site selection for cpEGFP insertion, to Supplemental Note1 and focus the main text on the protein engineering efforts that led to the sensors used in the final applications.

There are several aspects of the data that are missing or need clarification. Specifically, the authors do not report the quantum yield or the brightness

of the sensors, which has direct implications for how easy the sensors are to use.

Reply: We apologize for not reporting the quantum yield or brightness of the sensors. We understand the importance of this information for assessing the ease of use of the sensors. In the revised version, we provide the quantum yield data for the sensors to address this issue. Specifically, on line 312 of the manuscript, we added the following statement: "During the determination of quantum yield, we observed that the absolute quantum yield of the GECFINDER variants ranged from 0.3 to 0.6 upon addition of the respective substrates, which show no significant difference with that reported in literature (ref. 34). The specific values are provided in Extended Data Table 2." The detailed content of Extended Data Table 2 is as follows:

Extended Data Table2

Biosensor	Ligand	$\lambda_{\text{ex}}(\text{nm})$	$\lambda_{\text{em}}(\text{nm})$	ϕ^a	
				Apo	Sat
GECFINDER-4CA	4-coumaric acid	400	510	0.18	0.54
GECFINDER-Phe3	Phe	460	510	0.15	0.47
GECFINDER-Phe3.2	Phe	460	510	0.15	0.46
GECFINDER-Glu	Glu	480	510	0.41	0.56
GECFINDER-Pro	Pro	480	510	0.37	0.44
GECFINDER-Ile3	Ile	480	510	0.08	0.52

GECFINDER-His2	His	480	510	0.19	0.56
GECFINDER-Leu4	Leu	480	510	0.08	0.36
GECFINDER-S β F	S- β -Phe	460	510	0.32	0.37
GECFINDER-Tyr	Tyr	480	510	0.10	0.38

^a, Quantum Yield in the absence (Apo) or presence (Sat) of the corresponding analyte (4-coumaric acid, 100 μ M. Phe, 100 μ M for GECFINDER-Phe3, 1mM for GECFINDER-Phe3.2. Glu, 1mM. Pro, 1mM. Ile, 1mM. His, 1mM. Leu, 1mM. S- β -Phe, 1mM. Tyr, 1mM.).

Additionally, since the sensors appear to complete half of their catalytic reaction, care should be taken in terms of using KD to describe their dose-response profiles to their ligands since the catalytic activity means that this is not a simple binding affinity. Did the dose-response profiles change over time? How long were the titrations incubated prior to measurement?

Reply: Thank you for your valuable feedback regarding the use of K_d to describe the dose-response profiles of the biosensors. We have carefully considered your suggestion and have made the necessary revisions in the revised manuscript. We have replaced the use of K_d with *EC50* (half-maximal effective concentration) to describe the dose-response profiles in the revised manuscript. *EC50* represents the concentration of ligand at which half of the maximum response is achieved.

Regarding the stability and time-dependence of the dose-response curve, the dose-response curve is plotted after the sensor response reaches a stable state. Typically, the dose-response curve of the sensor stabilizes within 5

minutes after the addition of the analyte and remains unchanged thereafter. Once the stable state is reached, there are no further changes observed over time. The measurements were initiated immediately after adding the substrate, and readings were taken every minute for a continuous period of 15 minutes.

We have incorporated this additional information regarding the determination of dose-response curves into the Methods section, specifically in lines 757-761. “The dose-response curve was determined immediately after the addition of the analyte. The total duration of the measurement was 15 minutes, with readings taken at one-minute intervals. The dose-response curve was plotted using the fluorescence values obtained after achieving stability in the fluorescence signal.” These modifications will provide a more comprehensive understanding of the experimental procedures and ensure transparency in our data analysis.

Lastly, reported spectra of the sensors and their description require further clarification. Table 1 reports the excitation maximum of Phe3 to be 460 nm, yet a visual inspection of the spectra doesn't support this.

Reply: Thank you for your valuable comments and suggestions. Regarding the excitation maximum of GECFINDER-Phe3, we agree that based on the fluorescence spectra, the optimal excitation wavelength appears to be around 480 nm. However, as indicated in Table 1, the maximum $\Delta F/F_0$ for

GECFINDER-Phe3 is observed at 460 nm. Therefore, we selected 460 nm as the excitation wavelength for GECFINDER-Phe3. To address this discrepancy, we have added relevant explanatory notes in the Table 1 of manuscript, clarifying the rationale behind our choice of excitation wavelength. Specifically, “The excitation wavelength of GECFINDER-Phe is selected at the maximum $\Delta F/F_0$, not the wavelength at the maximum fluorescence intensity.”

Table 1

Excitation wavelength (nm)	Fluorescence intensity (0.001 μ M Phe)	Fluorescence intensity (100 μ M Phe)	$\Delta F/F_0$
430	992.6667	1731.667	1.744459
435	758.3333	1755.333	2.314725
440	623	1868.333	2.99893
445	581.6667	2105	3.618911
450	623.6667	2459.667	3.94388
455	721.3333	2930.667	4.062847
460	814.6667	3327	4.083879
465	932.6667	3674	3.939242
470	1032.667	3999.333	3.872821
475	1138	4306.333	3.784124
480	1239	4692	3.786925
485	1334.667	4963.333	3.718781
490	1367.667	4941.333	3.612966

Furthermore, some sensors appear to be ratiometric in excitation (GECFINDER-His2, 4CA), but are described in a confusing fashion.

Reply: Thank you for your feedback regarding the description of ratiometric sensors. We apologize for any confusion caused by the initial presentation of the data. We have revised the description of the excitation spectra for GECFINDER-4CA, and the updated details can be found in the manuscript from lines 196 to 200. Specifically, “ However, the GECFINDER-4CA was ratiometric in excitation. The emission spectrum shown in Fig.3 b was measured when the excitation wavelength was 450 nm. As a result, it exhibited an opposite trend compared to the dose–response curve depicted in Fig. 3 a.” We have provided a clearer and more coherent explanation to ensure a better understanding of the ratiometric behavior of GECFINDER-4CA.

The discussion is lacking in perspective that provides the reader with context to evaluate the sensors and their potential impact. For example, all of the demonstrations of the use of these sensor use them as purified reagents at relatively high concentrations. Yet, the authors suggest that in the future these sensors might be used to measure metabolites intracellularly. Not only were the sensors not demonstrated to be compatible with in cellulo applications, but a sensor for metabolites that consumes ATP seems disadvantageous for capturing metabolic dynamics. It would also be helpful to contextualize the use of a fluorescent protein,

rather than a system that might allow for amplification or provide more photons.

Reply: Thank you for raising your concerns regarding the compatibility of our sensors for intracellular applications and the limitations of using fluorescent proteins. We acknowledge your reservations about the application of these sensors within cells, and we agree that our current study has not demonstrated their suitability for intracellular use. In the revised version, we explicitly state this limitation and propose future research directions to further explore the applicability of the sensors for intracellular measurements. We focus on discussing the technical challenges associated with intracellular applications and provide potential solutions. For more detailed information, please refer to lines 630 to 642 of the manuscript. Specifically, “The intracellular application of fluorescent biosensors offers exciting possibilities for studying metabolic dynamics within living cells. While our current study focuses on characterizing and evaluating GECFINDER as purified reagents, the potential for intracellular use remains an area of future exploration. One potential key challenge in the intracellular application of GECFINDER is its ATP dependency, which can potentially impact the cellular environment and limit its utility. In general, intracellular ATP concentrations in animal and microbial cells range from 2.74 to 7.47 mM (ref. 54), providing sufficient ATP for GECFINDER. Moreover, due to the insertion of

cpEGFP, GECFINDER only undergoes a partial reaction, and the labile intermediate cannot be effectively released, resulting in a significantly slower ATP consumption process. This extremely slow ATP consumption rate is unlikely to hinder the intracellular application of GECFINDER. Furthermore, we can further weaken or eliminate GECFINDER's dependence on ATP through protein engineering.”

Regarding your point about the limitations of using fluorescent proteins, we recognize that they have certain drawbacks. However, this research represents our initial work in developing sensors based on the ANL family, and fluorescent proteins is a simple and relatively easy solution and it show great results for our applications. Additionally, other systems such as bioluminescent systems, may have disadvantages such as potential interference in signal amplification or the requirement for additional substrates. Furthermore, the conformational changes in ANL enzymes may not be effectively transmitted to amplification systems. We also acknowledge the need to explore alternative reporting systems that can overcome the limitations of fluorescent proteins in terms of signal amplification and photon emission. In subsequent studies, we have already begun investigating other systems that can address these challenges and potentially provide improved performance.

Some small changes:

The acronym POTS needs to be defined in the text

Reply: Thank you for pointing out the missing definition of the acronym "POTS" in the manuscript. However, upon reviewing the manuscript, we could not find any instances where "POTS" is used as an abbreviation. We are uncertain if you might be referring to the abbreviation "POCT" instead. If "POCT" is the intended abbreviation, we provide a specific explanation at line 77-78 of the manuscript. Specifically, "However, such methods lack the detection throughput capacity and simplicity that are urgently needed in many applications, such as point-of-care testing (POCT) scenarios."

cpEGFP changed to cpFP in the middle of the text and CpFP in the legend of Figure 2. Please regularize.

Reply: Thank you for your suggestion. We apologize for the confusion caused by using both "cpEGFP" and "cpFP" interchangeably. We have made the necessary adjustments to regularize the abbreviation throughout the text. We have corrected the legend of Figure 2 to ensure consistency.

Reviewer #2:

Wang et al. apply the strategy of fusing circularly permuted GFP to proteins that undergo a ligand-induced conformational changes to the ANL superfamily of enzymes. The novelty and impact of the present manuscript is in the application of this strategy to enzymes and the demonstration of its broad applicability. Several biosensors could be generated using rational design and screening methods and applications in sensing metabolites in blood or in screening for overproducing strains for biotechnological applications is demonstrated in extensive work.

The manuscript is generally sound and well written, however there are a number of grammatical problems or typos, however mostly easy to understand and correct. Also the figures are of high quality.

Reply: We sincerely appreciate the reviewer's positive comments.

Major points

When trying to generate a GLU-sensor by inserting cpGFP into PpsA-GluA, the authors state that they generated a sensor that detected ATP, but not glutamate (Extended Fig. 1 a-c). Extended Fig. 1c indicates that this sensor would also detect Mg²⁺ ions. However, more importantly, the authors add Mg²⁺ and ATP to all assays. ATP is cosubstrate for the enzymatic reaction and it probably binds together with ATP. The authors do not describe to what extent the other sensors depend on the ATP and

Mg²⁺ concentrations. This is important information that should be presented. While this may not be so relevant for the in vitro applications described by the authors, it would be relevant for potential in vivo applications.

Reply: We appreciate your valuable suggestion. We agree that clarifying the sensor's dependence on ATP and Mg²⁺ concentrations is crucial. In the revised manuscript, we have provided a detailed description of the impact of ATP and Mg²⁺ on GECFINDER sensors. Please refer to lines 299-312 of the manuscript and Extended Data Fig. 4 for specific information regarding the ATP and Mg²⁺ dependency of the sensors. Specifically, “We also investigated the dependence of GECFINDERS on Mg²⁺ and ATP (Extended Data Fig. 4). All GECFINDERS exhibited no significant change in fluorescence intensity in the presence of Mg²⁺ alone. Among them, the fluorescence intensity of GECFINDER-4CA is slightly reduced upon the presence of ATP and Mg²⁺ under 400 nm excitation, while the addition of 4-coumaric acid enhances the fluorescence intensity (Extended Data Fig. 4 a). This phenomenon corroborates with the distinct conformational changes observed in Hinge A of Nt4CL2 (Mg²⁺+ATP) (PDB ID: 5BSM) and Nt4CL2 (Coumaroyl-AMP) (PDB ID: 5BST) reported in the literature (Extended Data Fig. 4 k), where the direction of motion in Hinge A differs (ref. 25). GECFINDER-Phe3/Pro/Ile3/Leu4 exhibited a slight increase in fluorescence intensity when both Mg²⁺ and ATP were present (Extended

Data Fig. 4 b, d, e, g). On the other hand, GECFINDER-Glu/His2/sβF/Tyr/Phe3.2 showed no significant change in fluorescence intensity when Mg^{2+} and ATP were added (Extended Data Fig. 4 c, f, h, i, j).” Although the use of GECFINDER relies on the presence of cofactors Mg^{2+} and ATP, their availability in the cellular environment is generally sufficient to support the functioning of GECFINDER. Therefore, the dependence on ATP and Mg^{2+} is unlikely to have a significant impact on the intracellular application of GECFINDER.

Extended Data Fig. 4 **Dependence of GECFINDERS on Mg²⁺ and ATP.** **a-j**, Fluorescence intensities of GECFINDERS in the absence of any substrate, in the presence of Mg²⁺ alone, ATP alone, both Mg²⁺ and ATP, Mg²⁺ and ATP with respective substrates. Mg²⁺ concentration was 2.5 mM, and ATP concentration was 1 mM. **a**, 100 μM 4-coumaric acid, **b**, 100 μM phenylalanine, **c**, 1 mM glutamine, **d**, 1 mM proline, **e**, 100 μM isoleucine, **f**, 1 mM histidine, **g**, 1 mM leucine, **h**, 1 mM S-β-phenylalanine, **i**, 1 mM tyrosine, **j**, 1 mM phenylalanine. **k**, Conformational change of Nt4CL2 at Hinge A. All data shown are means ± S.D. (n=3).

The authors provide no information as to how the dose-response curves were fitted to obtain the specified KD values. The authors need to specify the exact mathematical formula used for fitting and specify for each curve which parameters have been obtained. Also the software should be stated. Some dose-response curves appear as if they may be fitted by a simple sigmoidal curve without apparent cooperativity, such as Fig. 3a. In others the fluorescence changes take place over a wider concentration range, indicating that a Hill coefficient was probably used for fitting. Fig. 3a is a dose-response curve that likely does not support Kd fitting and this fluorescence change may be generated by other mechanisms than the ligand-induced conformational change. In curves extended Fig. 2 e and i as well as extended Fig. 3 a and g the fit curves quickly go into saturation (become horizontal) at higher substrate concentration and I really wonder which equilibrium binding model generates such a binding curve.

Reply: Thank you for bringing these concerns to our attention. We apologize for not providing detailed information regarding the fitting of dose-response curves. In the revised version, we have included the software, mathematical formula, and parameters used for fitting each curve. All dose-response curves were fitted using the five-parameter model (Hill 5), with the formula $y = A_{\min} + (A_{\max} - A_{\min}) / (1 + (x_0/x)^h)^s$, and the specific details can be found in Extended Data 4.

Regarding the cooperativity issues of Fig. 3a, we have carefully reviewed your observation regarding Fig. 3a and its apparent lack of cooperativity. However, we are not sure about what you specifically mean by "cooperativity" in the context of Fig. 3a. We are not clear if you are referring to cooperativity in the context of GECFINDER-4CA forming multimers with multiple 4-coumaric acid substrate binding pockets, as the ligand-binding domain Nt4CL2 of GECFINDER-4CA is a monomer (ref. 25 of the manuscript), therefore it does not have cooperativity of multimeric state. If you are referring to cooperativity between 4-coumaric acid, ATP, and Mg^{2+} , we want to clarify that during the dose-response curve measurements, we performed the measurement with 2.5 mM Mg^{2+} and 1 mM ATP, which are under saturation level, before adding 4-coumaric acid. As a result, GECFINDER-4CA does not exhibit cooperativity between 4-coumaric acid and Mg^{2+} or ATP. We would appreciate it if you could kindly provide further clarification.

Regarding the curves in Extended Fig. 2e, 2i, and Extended Fig. 3a, 3g, where the fit curves quickly reach saturation at higher substrate concentrations. We speculate that this might be due to larger errors in the data points at the end of the dose-response curves. On the other hand, slightly altered pH values in the reaction system due to the higher substrate concentrations could lead to small variations in GECFINDER's fluorescence performance. Therefore, these factors may have contributed

to the fit curves quickly reaching saturation at higher substrate concentrations and then drop a little at end.

Minor points

Concerning the ANL enzymes used to design the biosensors. I would suggest to avoid the term biocatalyst and only use the term enzyme instead. "Biocatalyst" is usually (admittedly not always) used when referring to enzymes in a biotechnological sense for synthesis etc.

Reply: Thank you for your suggestion. We have made the appropriate adjustments in the manuscript by avoiding the use of the term 'biocatalyst' and using the term 'enzyme' instead to describe the ANL enzymes employed in our biosensor design.

On page 4, line 73 the authors cite two papers to state that cpFP-based biosensors usually have wider dynamic ranges as FRET-based biosensors. Please check if the cited papers really describe a comparison of FRET-based biosensors and cpFP and if you want to confirm this statement. For a non-expert, one should better specify "...usually have wider dynamic ranges (concerning the fluorescence change) as..." as one might apply "dynamic range" also to the analyte concentration.

Reply: Thank you for your suggestion. We have carefully reviewed the two cited papers to confirm that they indeed describe a comparison

between FRET-based biosensors and cpFP-based biosensors. In reference 13, it is stated, 'Compared with the FRET-based sensors which are ratiometric, single-FP-based intensimetric metal sensors have broader dynamic range and narrower excitation/emission wavelengths.' In reference 14, it is mentioned, 'Next, we intend to highlight the main advantages and disadvantages of cpFP-based probes over their counterparts, mainly FRET-based probes. Their most important benefit is the high dynamic range attributed to the efficient conformational coupling between sensory and reporter units, as described above. Moreover, even proteins with relatively modest conformational rearrangements can be tested as sensory units, providing an affordable signal-to-noise ratio for measurements. Second, they occupy a narrower part of the light spectrum, facilitating multiparameter imaging when several biochemical events are simultaneously monitored in a single living system.'

We believe that both of these papers indicate that cpFP-based biosensors have a wider dynamic range and occupy a narrower part of the light spectrum compared to FRET-based biosensors. However, we agree with your suggestion to clarify that the wider dynamic range refers to the fluorescence change rather than the analyte concentration. In the revised manuscript, we have specifically emphasized that the wider dynamic range pertains to the fluorescence change. Thank you for bringing this to our attention, and we appreciate your careful review of the cited literature.

Page 4, line 80: "Catalytic enzymes..." Enzymes are always catalytic

Reply: Thank you for your suggestion. We acknowledge that enzymes are inherently catalytic. We have removed the redundant term "catalytic" in the mentioned sentence on page 5, line 99.

Fig.1 is a nice overview figure. I wonder if the authors could replace the figure of the instrument in "GECFINDER screening" by a scheme that illustrates the principle of the screening process.

Reply: We appreciate your suggestion, and we have revised the figure accordingly to provide a clearer representation of the screening principle.

Please refer to the figure below for the specific details:

Fig. 1 The GECFINDER screening method overview. The GECFINDER screening method can be roughly divided into 4 parts. The first part is the design of GECFINDER. After obtaining the ANL superfamily protein sequences, if

there is no crystal structure, homology modeling is needed to determine the positions of hinge A and hinge B according to its three-dimensional structure, and cpEGFP with random linker is inserted into hinge A or hinge B. The second part is the construction of GECFINDER random linker library. Firstly, the DNA sequence of ANL superfamily member is integrated into the expression plasmid, and then random linker primers are designed at the insertion position of cpEGFP, and the linear DNA fragment of LBD is obtained by PCR. The cpEGFP DNA fragment is integrated into the linear DNA fragments of LBD. The plasmid of GECFINDER random linker library is transferred into *Escherichia coli* to induce expression of GECFINDER random linker library. The third part is the screening of GECFINDER random linker library. The differences in fluorescence values of GECFINDER random linker library before and after the addition of ligand are detected by the microplate reader, and the effective GECFINDER is cultured, purified, and characterized. The fourth part is the various potential applications of GECFINDER including FADS screening, tissue cell imaging, POCT assay, etc.

Page 12, line 223: "By comparing the sequences of the GrsA-PheA with a known structure (PDB ID: 223 1AMU) and the By comparing the sequences other 160 A domains, the...": Replace "and the other" by "with".

Reply: We gratefully appreciate for your valuable comment. We have revised the sentence on page 13, line 240 to replace "and the other" with "with" for clarity and accuracy.

In Table 2, "ND" is specified for the operating range and KD for GECFINDER-Leu1 and Ligand Phe. Explain if ND stands for not determined or not detectable and why this is the case, although $\Delta F/F_0$ is reported.

Reply: We apologize for the confusion caused by the "ND" specification in the operating range and KD value for GECFINDER-Leu1. GECFINDER-Leu1 showed an increase in fluorescence intensity upon the addition of Phe; however, the resulting $\Delta F/F_0$ values were relatively low, leading to a low signal-to-noise ratio for the sensor. Consequently, fitting

dose-response curves under such conditions would result in significant errors, making the determination of accurate operational range and KD values unreliable. Hence, we did not provide the operational range and KD value for GECFINDER-Leu1. We have now added relevant clarification to Table 2 in the manuscript. Specifically, “Due to the $\Delta F/F_0$ values of GECFINDER-Leu1 for Phe being less than 0.26, the signal-to-noise ratio was low, resulting in significant errors when fitting dose-response curves. As a result, the operating range and $EC50$ values for this sensor was not provided in this study.”

Reviewer #3:

The manuscript from Wang et al., shows the repurposing of a common biosynthetic enzyme fold for a new utility as a biosensor. Specifically, the authors focus on enzymes of the ANL superfamily, which are known to undergo conformational rotation during the catalytic cycle. Insertion of circularly permuted GFP superfolder into the loop region between the adenylation N domain and the C domain or between the adenylation domain and carrier domain yields an enzyme that undergoes fluorescence shift in response to a substrate of choice. The idea is clever and may have some utility. While the current response requires low to high micromolar concentrations of substrate, this is something that can be improved upon in

future iterations. Overall I am enthusiastic about this work but there are several points that need to be clarified in a revised version. Specifically,

Reply: We sincerely appreciate the reviewer's positive comments.

1. I am confused why the repurposed catalysts do not turn over. The dose responses only show linear increases. The authors should clarify or provide a rationale for why this is so.

Reply: We appreciate your inquiry regarding the absence of turnover in the ANL enzyme members utilized in our study. The ANL enzymes perform substrate binding and activation through a two-step catalytic reaction. In the first half-reaction, the substrate enters the binding pocket and consumes ATP to form a substrate-AMP intermediate. Insertion of cpEGFP into Hinge A or Hinge B does not significantly affect the catalytic performance of the N-terminal large subunit, probably allowing the ANL enzymes' first half-reaction to proceed largely unaffected in the GECFINDER system.

However, when cpEGFP is inserted into Hinge A, the second half-reaction, which involves the C-terminal small subunit's active site replacing AMP with CoA, is significantly inhibited due to the altered spatial positioning between the N-terminal large subunit and C-terminal small subunit caused by cpEGFP insertion. As a result, turnover is not observed. Similarly, when cpEGFP is inserted into Hinge B, the relative spatial positioning between the PCP domain and A domain is affected, preventing the transfer of the

substrate-AMP complex to the PCP domain, thus resulting in the absence of turnover. It is the insertion of cpEGFP that hinders the turnover process in GECFINDER, leading to a linear segment in the dose-response curve. We hope this explanation clarifies the absence of turnover in the ANL enzymes and provides insight into the linear increases observed in the dose-response curves. Thank you for raising these questions, and if you have any further inquiries or concerns, please do not hesitate to let us know.

2. Figure 2 is entirely too confusing as drawn. Not all ANL enzymes have a PCP domain so the top half of the panels do not make sense in the context of panel A.

Reply: We appreciate your feedback regarding the clarity of Figure 2. In the revised manuscript, we have addressed this concern by removing the content of Figure 2a and transferring the information it contained to Figures 2b and 2c. The updated Figure 2 is shown below:

Fig. 2 Mechanism illustration of GECFINDER. **a and b**, Overview of the reactions catalyzed by acyl- and aryl-CoA synthetases and NRPS adenylation domains of the ANL superfamily. The hinge between the N-terminal large subunit and the C-terminal small subunit in the ANL superfamily is called hinge A, and the hinge between the C-terminal small subunit and the PCP domain is called hinge B. **c**, cpEGFP is inserted into the hinge A between the N-terminal large subunit and the C-terminal small subunit of ANL superfamily members. When GECFINDER binds to its ligands and initiates the adenylation reaction, the N-terminal large subunit and C-terminal small subunit are close to each other, resulting in a change in the fluorescence intensity of cpEGFP. **d**, cpEGFP is inserted into the end of the C-terminal small subunit of ANL superfamily members. When GECFINDER binds to its ligands and initiates the adenylation reaction, the N-terminal large subunit and C-terminal small subunit are close to each other, and the conformational changes at the C-terminal small subunit induces changes in the fluorescence intensity of cpEGFP. **e**, cpEGFP is inserted into hinge B between the C-terminal small subunit of the ANL superfamily member and the PCP domain. When GECFINDER binds to its ligands and catalyzes the thioesterification reaction, the PCP domain is close to the C-terminal small subunit, and the conformational change of the c-terminal small subunit and the PCP domain leads to the change of the fluorescence intensity of cpEGFP.

3. There should be a more robust discussion of signal/noise of the response, especially if the authors are going to propose this as a system for use in biology.

Reply: We gratefully appreciate for your valuable comment. In the revised manuscript, we have expanded the discussion on signal-to-noise ratio,

emphasizing the importance of this parameter for the practical application of the biosensors. Specifically, in lines 643-646 of the manuscript, we have added the following content: "Additionally, achieving a high signal-to-noise ratio poses another challenge for intracellular application. Although some GECFINDER variants (Ile, His, Pro) currently exhibit lower signal-to-noise ratios, we have shown that protein engineering can rapidly improve the performance of various GECFINDERS." We have also included considerations for optimizing the signal-to-noise ratio in future studies.

4. As I understand it, all of these engineered ANL enzymes require a concentration of ATP to maintain saturation. This is only briefly mentioned in the manuscript and should be stated more explicit. Moreover, where possible, the known literature K_d values for ATP for the various systems under study should also be stated.

Reply: We appreciate your observation regarding the ATP concentration dependency in the engineered ANL enzymes. We have found literature-reported K_m values for ATP in GrsA A domain and At4CL2. In the revised manuscript, we have included this information and provided additional details in lines 387-390. Specifically, "While we did not find specific information on the K_m value of ATP for Nt4CL2, a closely related homolog, At4CL2, has a reported ATP K_m value of 0.163 mM (ref. 35).

Additionally, the GrsA A domain exhibits a K_m value of 0.15 mM for ATP (ref. 36).” However, different cell types or cellular conditions may exhibit variations in ATP concentrations, which could impact GECFINDER's performance. Therefore, the potential limitations imposed by ATP availability should be carefully considered when applying GECFINDER in different cellular contexts.

REVIEWER COMMENTS

Reviewer #1 (Remarks to the Author):

The proposed revisions largely address many of the concerns of myself and my fellow reviewers and I commend the authors for the work required to do so.

There are still a couple of small issues that I think merit mentioning. EC50 is a more appropriate metric for these sensors, but I would caution the authors that it only represents the dose-response profile of a sensor (amino acid to fluorescence), not its thermodynamic affinity. Regarding the discussion raised by another reviewer, I agree that the dose-response curve fitting should be more explicit, and I cannot find any addition in the methods that details the equation and how it was fit as was requested by the other reviewer. I also see the differences in the curves, which are likely related to different Hill coefficients being required to fit the data (i.e. different apparent cooperativity). Here cooperativity relates to the need for a coefficient and makes no assumption about the underlying biochemical mechanism.

There are still instances where cpFP has not be replaced by cp(E)GFP: Line 316, 337 (and the whole section thereafter), 443, 653 (and following paragraph).

Please state the excitation wavelengths used for the quantum yield measurements.

I would suggest moderating the statement at line 396 as there are several non-transcription factor biosensors for amino acids that exhibit dF/F that are much higher than those detailed in this report, and which could easily be used for these types of applications.

Reviewer #2 (Remarks to the Author):

The authors have addressed and clarified the points raised in my review. There is only one point left, that needs to be addressed by the authors. It concerns the function used to fit the dose-response curves. The authors have now specified the fit function (HILL5 in Origin 2021) and the values of the fitted parameters in Extended Data 4. HILL5 appears to correspond to LOGISTIC5 in the current Origin

documentation found at <https://www.originlab.com/doc/Origin-Help/Logistic5-FitFunc>. I have not seen this fitting equation, which includes a control parameter s , being applied to dose-response curves. The authors should describe why a classical fit with a Hill coefficient (without s) was not sufficient and appropriate to analyze the data. Inclusion of the parameter s likely results in the "unusual" (or perhaps even "unlikely") fit curves, in which the binding curve "suddenly" goes into saturation, for example in Extended Data Fig. 3 a and g. This might result in lower EC50 values compared to the classical Hill model of cooperative binding. However, it is unlikely to result in large changes affecting the interpretation of the data.

My remarks in the first review concerning cooperativity were related to the slope of the dose-response curve in terms of apparent ligand binding cooperativity.

Reviewer #3 (Remarks to the Author):

The authors have addressed all of the concerns raised during the review of the initial submission. I don't have anything more to add and I hope to see this work in print soon.

Reviewers' comments & the authors' replies

Reviewer #1:

The proposed revisions largely address many of the concerns of myself and my fellow reviewers and I commend the authors for the work required to do so.

Reply: We sincerely appreciate the reviewer's positive comments.

There are still a couple of small issues that I think merit mentioning. EC50 is a more appropriate metric for these sensors, but I would caution the authors that it only represents the dose-response profile of a sensor (amino acid to fluorescence), not its thermodynamic affinity.

Reply: We appreciate your feedback. We completely understand your description of EC50 as a measure of the dose-response relationship not the thermodynamic affinity of the sensor. In our manuscript, we aimed to present the EC50 values as an indication of the sensors' sensitivity to the substrate rather than a direct representation of thermodynamic affinity.

Regarding the discussion raised by another reviewer, I agree that the dose-response curve fitting should be more explicit, and I cannot find any addition in the methods that details the equation and how it was fit as was requested by the other reviewer. I also see the differences in the curves, which are likely related to different Hill coefficients being required to fit the data (i.e. different apparent cooperativity). Here cooperativity relates

to the need for a coefficient and makes no assumption about the underlying biochemical mechanism.

Reply: Thank you for your thoughtful feedback. We apologize for any confusion caused by the lack of explicit details regarding the dose-response curve fitting in our methods. We understand that the term "cooperativity" in this context doesn't assume any specific biochemical mechanism but rather emphasizes the requirement for a coefficient to describe the observed dose-response relationship. In response to your comments, we provided a more comprehensive explanation of the curve fitting process, including the equation used and the rationale behind selecting specific parameters. For specific details, please refer to our response to Reviewer 2's query regarding the HILL model.

There are still instances where cpFP has not be replaced by cp(E)GFP: Line 316, 337 (and the whole section thereafter), 443, 653 (and following paragraph).

Reply: Thank you for your suggestion. The term "cpFP" refers to "circularly permuted fluorescent protein." We have used this term to generally refer to biosensors based on circularly permuted fluorescent proteins. In instances where we discuss biosensors based on circularly permuted fluorescent proteins in a broad sense, we have retained the term "cpFP" for its inclusivity. However, in reference to our specific research, we have diligently replaced all instances of "cpFP" with "cpEGFP."

Please state the excitation wavelengths used for the quantum yield measurements.

Reply: Thank you for your suggestion. The excitation wavelengths used for the quantum yield measurements have been provided in Extended Data Table 2.

Extended Data Table 2

Biosensor	Ligand	$\lambda_{ex}(nm)$	$\lambda_{em}(nm)$	ϕ^a	
				Apo	Sat
GECFINDER-4CA	4-coumaric acid	400	510	0.18	0.54
GECFINDER-Phe3	Phe	460	510	0.15	0.47
GECFINDER-Phe3.2	Phe	460	510	0.15	0.46
GECFINDER-Glu	Glu	480	510	0.41	0.56
GECFINDER-Pro	Pro	480	510	0.37	0.44
GECFINDER-Ile3	Ile	480	510	0.08	0.52
GECFINDER-His2	His	480	510	0.19	0.56
GECFINDER-Leu4	Leu	480	510	0.08	0.36
GECFINDER-S β F	S- β -Phe	460	510	0.32	0.37
GECFINDER-Tyr	Tyr	480	510	0.10	0.38

^a, Quantum Yield in the absence (Apo) or presence (Sat) of the corresponding analyte (4-coumaric acid, 100 μ M. Phe, 100 μ M for GECFINDER-Phe3, 1mM for GECFINDER-Phe3.2. Glu, 1mM. Pro, 1mM. Ile, 1mM. His, 1mM. Leu, 1mM. S- β -Phe, 1mM. Tyr, 1mM.).

I would suggest moderating the statement at line 396 as there are several non-transcription factor biosensors for amino acids that exhibit dF/F that are much higher than those detailed in this report, and which could easily be used for these types of applications.

Reply: Thank you for your valuable suggestion. Upon careful consideration, we recognize the need to moderate the statement at line 396 for accuracy and clarity. The revised statement now reads as follows: "Transcription factor biosensors can be used as screening tools for high-throughput screening of amino acids producing strains (ref.38), but their application for rapid and accurate in vitro quantification of amino acids is less common."

Reviewer #2:

The authors have addressed and clarified the points raised in my review.

Reply: We sincerely appreciate the reviewer's positive comments.

There is only one point left, that needs to be addressed by the authors. It concerns the function used to fit the dose-response curves. The authors have now specified the fit function (HILL5 in Origin 2021) and the values of the fitted parameters in Extended Data 4. HILL5 appears to correspond to LOGISTIC5 in the current Origin documentation found at <https://www.originlab.com/doc/Origin-Help/Logistic5-FitFunc>. I have not seen this fitting equation, which includes a control parameter s , being

applied to dose-response curves. The authors should describe why a classical fit with a Hill coefficient (without s) was not sufficient and appropriate to analyze the data. Inclusion of the parameter s likely results in the "unusual" (or perhaps even "unlikely") fit curves, in which the binding curve "suddenly" goes into saturation, for example in Extended Data Fig. 3 a and g. This might result in lower EC50 values compared to the classical Hill model of cooperative binding. However, it is unlikely to result in large changes affecting the interpretation of the data.

Reply: Thank you for your valuable input and for highlighting a critical point concerning our chosen approach for fitting the dose-response curves. We greatly appreciate your meticulous examination of the Hill5 (LOGISTIC5) model and its potential implications on our data analysis. In general, the fitting of dose-response curves often employs the classical Hill4 model, which assumes symmetry of dose-response curves around the midpoint¹. However, some dose-response curves exhibit asymmetry that can lead to inaccurate EC50 estimations when fitted using the Hill4 model². To address this limitation, Van der Graaf and Schoemaker introduced the Richards equation, also known as the five-parameter logistic equation or Hill5 model, which offers more comprehensive fitting for asymmetric dose-response data³. The Hill4 model, sometimes referred to as the four-parameter logistic equation, fits the curve's bottom, top, EC50, and Hill slope. Hill5 introduces an additional parameter " s " to quantify the observed

asymmetry. When "s" equals 1, the Richards equation converges with the Hill equation (Hill4), producing symmetric curves. However, when "s" \neq 1, the curves become asymmetric.

The Hill4 equation is a simpler version of the Hill5 equation, implying that these two models are nested. The data can be fit to both models, followed by an extra F-test to compare the fitting outcomes². The result of this model comparison helps determine whether a symmetric or asymmetric model better describes the data.

The statistical significance of whether a model with more parameters significantly improves the assumptions of a nested model with fewer parameters can be assessed using the F statistic⁴, defined as:

$$F = ((SS1 - SS2) / (df1 - df2)) / (SS2 / df2)$$

where SS represents the sum of squared residuals, df indicates the degrees of freedom, and the subscripts 1 and 2 correspond to models with fewer and more parameters, respectively. Generally, if the computed F value surpasses a critical threshold, it signifies that the inclusion of the extra parameter (in this case, parameter "s") substantially enhances the model's fit. The choice of the critical value is often linked to the selected significance level, commonly 0.05 or 0.01. Therefore, if the calculated F value significantly exceeds the critical value, it can be concluded that the Hill5 model, relative to the Hill4 model, performs better in interpreting the data, and this improvement is statistically significant. However, if the F

value is not significant, even with the introduction of more parameters in the Hill5 model, it cannot definitively prove its superiority in terms of fitting over the Hill4 model. The critical F values can be obtained by referencing an F-distribution table or by using statistical software. In software such as Excel, the F.INV() function can compute the critical F value. Computed for a significance level of 0.01, the critical F value for degrees of freedom 8 and 7 is 0.162, and for a significance level of 0.05, it's 0.286.

In light of this, the ongoing debate in our paper regarding dose-response curve fitting models centers around Extended Data Fig. 2 e, i and Extended Data Fig. 3 a-phe, g. These four curves exhibit a sudden flattening at the end. We have now fit these four curves and the normally shaped Fig. 3a using both the Hill4 and Hill5 models. Subsequently, an F-test was conducted to determine the more appropriate model for each case. The degrees of freedom, sum of squared residuals, and computed F-values for these five dose-response curves within both Hill4 and Hill5 models are listed in Table 1.

Table 1

	Hill4		Hill5		F-value
	Degrees of Freedom	Residual Sum of Squares	Degrees of Freedom	Residual Sum of Squares	
Extended Data Fig. 2 e	8	4.97	7	2.69	5.93
Extended Data Fig. 2 i	8	21.27	7	18.58	1.01
Extended Data Fig. 3 a-Phe	8	38.60	7	32.01	1.44
Extended Data Fig. 3 g	8	5.36	7	3.88	2.67
Fig.3 a	8	7.89	7	6.54	1.44

At a significance level of 0.01, the computed critical F value is 0.162, and at a significance level of 0.05, it's 0.286. This implies that the computed F values in Table 1 are all greater than 0.162 and 0.286. Consequently, at both the 0.01 and 0.05 significance levels, it can be inferred that the Hill5 model outperforms the Hill4 model. This outcome suggests that the introduction of the additional parameter "s" significantly enhances the model's fit, leading to a better performance of the Hill5 model in interpreting the data. This underscores the necessity of introducing parameter "s" to quantitate asymmetry in the dose-response curve fitting of biosensors.

We hope this comprehensive explanation clarifies the rationale behind our choice of the Hill5 model and its statistical validation. If you have any further inquiries or concerns, please do not hesitate to reach out.

My remarks in the first review concerning cooperativity were related to the slope of the dose-response curve in terms of apparent ligand binding cooperativity.

Reply: Thank you for your feedback. In the dose-response curve of GEFINDER-4CA, the slope is 0.86. If the slope is greater than 1, it indicates apparent ligand binding cooperativity. However, with a slope less than 1 in the dose-response curve of GEFINDER-4CA, it suggests the absence of apparent ligand binding cooperativity.

Reviewer #3:

The authors have addressed all of the concerns raised during the review of the initial submission. I don't have anything more to add and I hope to see this work in print soon.

Reply: We sincerely appreciate the reviewer's positive comments.

References

1. Gottschalk PG, Dunn JR. The five-parameter logistic: a characterization and comparison with the four-parameter logistic. *Analytical biochemistry* **343**, 54-65 (2005).
2. Motulsky H, Christopoulos A. *Fitting models to biological data using linear and nonlinear regression: a practical guide to curve fitting*. Oxford University Press (2004).
3. Van der Graaf P, Schoemaker R. Analysis of asymmetry of agonist concentration–effect curves. *Journal of pharmacological and toxicological methods* **41**, 107-115 (1999).
4. Giraldo J, Vivas NM, Vila E, Badia A. Assessing the (a) symmetry of concentration-effect curves: empirical versus mechanistic models. *Pharmacology & therapeutics* **95**, 21-45 (2002).

REVIEWERS' COMMENTS

Reviewer #2 (Remarks to the Author):

The authors have fully addressed my comments concerning the models to fit the dose-response-curves. This point is now also documented in the methods part of the main manuscript text and in the peer review files.

I sincerely thank the authors for the detailed and thorough information provided in the reply to the reviewers comments. This was very helpful.

Reviewers' comments & the authors' replies

Reviewer #2:

The authors have fully addressed my comments concerning the models to fit the dose-response-curves. This point is now also documented in the methods part of the main manuscript text and in the peer review files.

I sincerely thank the authors for the detailed and thorough information provided in the reply to the reviewers comments. This was very helpful.

Reply: We sincerely appreciate the reviewer's positive comments.